# RIFT: Group-Relative RL Fine-Tuning for Realistic and Controllable Traffic Simulation

## Abstract

Achieving both realism and controllability in closed-loop traffic simulation remains a key challenge in autonomous driving. Dataset-based methods reproduce realistic trajectories but suffer from *covariate shift* in closed-loop deployment, compounded by simplified dynamics models that further reduce reliability. Conversely, physics-based simulation methods enhance reliable and controllable closed-loop interactions but often lack expert demonstrations, compromising realism. To address these challenges, we introduce a dual-stage AV-centric simulation framework that conducts imitation learning pre-training in a data-driven simulator to capture trajectory-level realism and route-level controllability, followed by reinforcement learning fine-tuning in a physics-based simulator to enhance style-level controllability and mitigate covariate shift. In the fine-tuning stage, we propose *RIFT*, a novel group-relative RL fine-tuning strategy that evaluates all candidate modalities through group-relative formulation and employs a surrogate objective for stable optimization, enhancing style-level controllability and mitigating covariate shift while preserving the trajectory-level realism and route-level controllability inherited from IL pre-training. Extensive experiments demonstrate that *RIFT* improves realism and controllability in traffic simulation while simultaneously exposing the limitations of modern AV systems in closed-loop evaluation.

## 1 Introduction

Reliable closed-loop traffic simulation is critical for developing advanced autonomous vehicle (AV) systems, supporting training and evaluation Feng et al. (2023b); Ding et al. (2023). An ideal traffic simulation should possess two key properties: *realistic*, reflecting real-world driving behavior; *controllable*, enabling customizable traffic simulation according to user requirements.

To balance these two essential properties, existing traffic simulation methods adopt different trade-offs depending on the underlying platform, often favoring either realism or controllability, as illustrated in Figure 1. Methods based on data-driven simulators exploit real-world data to generate realistic trajectories by learning multimodal behavioral patterns through imitation learning (IL) Ngiam et al. (2021); Sun et al. (2022); Feng et al. (2023a); Mahjourian et al. (2024). In addition to realism, recent studies on data-driven simulators have pursued controllability by conditioning scenario generation on user-specified inputs—such as text conditions Zhang et al. (2024); Tan et al. (2023), goal conditions Tan et al. (2024); Rowe et al. (2024), or cost functions Zhong et al. (2023b); Jiang et al. (2023b); Zhong et al. (2023a)—producing scenarios that are both realistic and aligned with user requirements. However, their open-loop training paradigm introduces the *covariate shift* problem during closed-loop deployment, arising from the distribution mismatch between training and deployment states. Moreover, data-driven simulators often adopt simplified environment dynamics Gulino et al. (2023); Caesar et al. (2021), resulting in unrealistic interactions and state transitions that further degrade closed-loop reliability. In contrast, physics-based simulators provide fine-grained control over scenario configuration through physical engines, enabling high-fidelity closed-loop interactions. Nonetheless, the absence of expert demonstrations makes it challenging to reproduce realistic behavior. To mitigate this, several approaches employ reinforcement learning (RL) to directly acquire controllable behaviors through interaction with the simulator Ding et al. (2021); Hanselmann et al. (2022); Chen et al. (2024b); Zhang et al. (2023b), although often at the cost of realism. Other approaches enhance realism by injecting real-world traffic data into physics-based simulators Osiński et al. (2020); Li et al. (2023), but typically rely on log-replay or rule-based simulation, limiting

Figure 1: **Traffic Simulation across Different Platforms.** (a) Data-driven Simulator: employs imitation learning to replicate real-world driving behaviors, but suffers from covariate shift and simplified dynamics; (b) Physics-based Simulator: enables controllable scenario construction via high-fidelity closed-loop interaction, but lacks large-scale real-world data; (c) Our framework: combines IL pre-training in a data-driven simulator to ensure realism with RL fine-tuning in a physics-based simulator to enhance controllability.

controllability and interactivity. Despite recent advances, a fundamental trade-off persists between realism and controllability across both paradigms, making it challenging to achieve both simultaneously in interactive closed-loop scenarios.

Drawing inspiration from the widely adopted "pre-training and fine-tuning" paradigm in large language models (LLMs) Rafailov et al. (2023); Yu et al. (2025); Shao et al. (2024), we combine the strengths of two platforms. Specifically, we perform IL pre-training in a data-driven simulator to capture realism, followed by RL fine-tuning in a physics-based simulator to address covariate shift and enhance controllability.

Building on this insight, we propose a dual-stage AV-centric simulation framework (Figure 1) that unifies the strengths of data-driven and physics-based simulators through a "pre-training and fine-tuning" paradigm, balancing realism and controllability in traffic simulation. In Stage 1, we pre-train a planning model via IL to generate realistic and multimodal trajectories conditioned on given route-level reference lines. This stage achieves both trajectory-level realism, capturing realistic and multimodal behavior patterns, and route-level controllability, guaranteeing compliance with prescribed reference lines. In Stage 2, we identify critical background vehicles (CBVs) through route-level interaction analysis, focusing on those most likely to interact with the AV. For these CBVs, we leverage the IL pre-trained model from Stage 1, conditioned on their route-level reference lines, to automatically generate realistic and multimodal trajectories that remain route-level controllable. On top of these generated candidates, we introduce *RIFT*, a novel group-relative RL fine-tuning strategy that improves controllability over driving styles and mitigates covariate shift. Unlike prior methods Zhang et al. (2023a); Peng et al. (2024) that fine-tune only the best trajectory or action, *RIFT* evaluates all candidate modalities via group-relative formulation Shao et al. (2024) and employs a surrogate objective for stable optimization, enhancing style-level controllability and alleviating covariate shift while preserving the trajectory-level realism and route-level controllability established in Stage 1.

Our contributions can be summarized as:

- We propose a dual-stage AV-centric simulation framework that combines IL pre-training in a data-driven simulator and RL fine-tuning in a physics-based simulator, leveraging their complementary strengths to balance realism and controllability.

- We propose *RIFT*, a novel group-relative RL fine-tuning strategy that evaluates all candidate modalities through group-relative formulation and employs a surrogate objective for stable optimization, improving style-level controllability and alleviating covariate shift, while retaining the trajectory-level realism and route-level controllability inherited from IL pre-training.

- Extensive experiments demonstrate that *RIFT* enhances the realism and controllability of traffic simulation, effectively exposing the limitations of modern AV systems under closed-loop settings.

## 2 RELATED WORK

**Realistic Traffic Simulation.** A variety of generative architectures have been explored for realistic traffic simulation Tan et al. (2021); Zhang et al. (2023c); Yang et al. (2025), including conditional variational autoencoders Suo et al. (2021); Rempe et al. (2022); Xu et al. (2023), diffusion-based models Jiang et al. (2024); Chitta et al. (2024); Zhou et al. (2025); Tan et al. (2025) and **GAIL-based approaches** Kuefler et al. (2017); Bhattacharyya et al. (2022); Chen et al. (2022). However, maintaining long-term stability remains challenging due to the *covariate shift* between open-loop training and closed-loop deployment. Recent methods such as SMART Wu et al. (2024), GUMP Hu et al. (2024), Trajeglish Philion et al. (2023), **LLMAD** Wang et al. (2025), **RLFTSim** Ahmadi & Schofield, and MotionLM Seff et al. (2023) address this issue by formulating traffic simulation as a next-token prediction (NTP) task, leveraging discrete action spaces to improve closed-loop robustness. Despite these advances, most approaches remain confined to data-driven simulation platforms Gulino et al. (2023); Caesar et al. (2021); Dauner et al. (2024); Montali et al. (2023), which typically adopt simplified environment dynamics. Such oversimplifications limit the reliability of long-term closed-loop interactions, especially in complex and interactive scenarios.

**Controllable Traffic Simulation.** Recent studies have introduced diverse conditioning mechanisms to generate traffic scenarios aligned with user preferences. CTG Zhong et al. (2023b) and MotionDiffuser Jiang et al. (2023b) employ diffusion models conditioned on cost-based signals. Language-conditioned methods, including CTG++ Zhong et al. (2023a), LCTGen Tan et al. (2023), and ProSim Tan et al. (2024), enable user specification through language prompts. Other strategies adopt guided sampling (SceneControl Lu et al. (2024)), retrieval-based generation (RealGen Ding et al. (2024)), or reward-driven causality modeling (CCDiff Lin et al. (2024)). Despite improving controllability, existing approaches remain confined to open-loop settings or simplified dynamics, and primarily target low-level control. High-level attributes such as driving style are underexplored, leaving the integration of realism and controllability in closed-loop simulation an open challenge.

**Closed-Loop Fine-Tuning.** Covariate shift—the mismatch between open-loop training and closed-loop deployment—remains a key challenge for reliable long-term traffic simulation. To address this, recent work explores fine-tuning strategies in the closed-loop setting. Hybrid IL and RL methods Zhang et al. (2023a); Peng et al. (2024); Lu et al. (2023) enhance robustness but typically fine-tune the entire model via RL, which often compromises realism due to the difficulty of designing human-aligned reward functions. **Gen-Drive** Huang et al. (2025b) **improves generative quality but does not optimize trajectory probabilities, making it inadequate for traffic simulation tasks that require faithful multimodal distributions.** Supervised fine-tuning approaches such as CAT-K Zhang et al. (2025) show strong performance but rely on expert demonstrations, limiting scalability. TrafficRLHF Cao et al. (2024) improves alignment through reinforcement learning with human feedback (RLHF), but demands costly human input and suffers from reward model instability. Moreover, most existing methods focus on optimizing the best action or trajectory, ignoring the inherent multimodality of traffic simulation, thus limiting behavioral diversity during fine-tuning.

## 3 BACKGROUND

### 3.1 TASK REDEFINITION

Following the widely adopted paradigm for closed-loop training and evaluation in autonomous driving Jia et al. (2024); Xu et al. (2022), our simulation framework includes a single autonomous vehicle (AV) navigating a predefined global route, accompanied by multiple rule-based background vehicles (BVs), forming an AV-centric closed-loop simulation environment. These BVs either provide diverse interactive data for training or serve to evaluate the AV's robustness. Building upon this setup, we identify a subset of critical background vehicles (CBVs) that are more likely to interact with the AV. For these CBVs, the rule-based control is replaced with a well-trained planning model, enabling the synthesis of realistic and controllable behaviors in interactive closed-loop scenarios.

### 3.2 CBV-CENTRIC REALISTIC TRAJECTORY GENERATION

With recent advances in imitation learning, data-driven approaches have demonstrated strong performance in generating realistic, multimodal trajectories Zheng et al. (2025); Huang et al. (2023);

Hu et al. (2023); Jiang et al. (2023a); Sun et al. (2024). In fully observable simulation environments, Pluto Cheng et al. (2024a) produces reliable, realistic, and multimodal trajectories by leveraging ground-truth states, while enabling route-level controllability through reference line encoding. These capabilities make Pluto a suitable choice for our planning model.

**CBV-Centric Scene Encoding.** Following Cheng et al. (2024a), for each CBV in the scene, we extract its current feature $F_{\mathrm{cbv}}$, the historical features of neighboring vehicles $F_{\mathrm{neighbor}}$, and vectorized map features $F_{\mathrm{map}}$. These features are encoded into $E_{\mathrm{cbv}} \in \mathbb{R}^{1 \times D}$, $E_{\mathrm{neighbor}} \in \mathbb{R}^{N_{\mathrm{neighbor}} \times D}$, and $E_{\mathrm{map}} \in \mathbb{R}^{N_{\mathrm{map}} \times D}$, respectively, where $N_{\mathrm{neighbor}}$ and $N_{\mathrm{map}}$ denote the number of neighboring vehicles and map elements, and $D$ is the embedding dimension. To model the interactions among these embeddings, we concatenate them and apply a global positional embedding (PE) to obtain the unified scene embedding $E_s \in \mathbb{R}^{(1+N_{\mathrm{neighbor}}+N_{\mathrm{map}}) \times D}$ as:

$$E_s = \mathrm{concat}(E_{\mathrm{cbv}}, E_{\mathrm{neighbor}}, E_{\mathrm{map}}) + \mathrm{PE}. \tag{1}$$

This scene embedding $E_s$ is then passed through $N$ Transformer encoder blocks for feature aggregation, yielding the final CBV-centric scene embedding $E_{\mathrm{enc}}$. Each encoder block follows the standard Transformer formulation. Specifically, the $i$-th block is defined as:

$$\begin{aligned} E_s^i &= E_s^{i-1} + \mathrm{MHA}(\mathrm{LayerNorm}(E_s^{i-1})), \\ E_s^i &= E_s^i + \mathrm{FFN}\left(\mathrm{LayerNorm}(E_s^i)\right), \end{aligned} \tag{2}$$

where MHA is the standard multi-head attention function, FFN is the feedforward network layer.

**Multimodal Trajectory Decoding.** To capture the multimodal nature of real-world driving behaviors, we adopt the longitudinal-lateral decoupling mechanism proposed in Cheng et al. (2024a). This approach leverages reference line information to construct high-level lateral queries $Q_{\mathrm{lat}} \in \mathbb{R}^{N_{\mathrm{ref}} \times D}$, and introduces learnable longitudinal queries $Q_{\mathrm{lon}} \in \mathbb{R}^{N_{\mathrm{lon}} \times D}$. These are concatenated and projected to form the multimodal navigation query $Q_{\mathrm{nav}} \in \mathbb{R}^{N_{\mathrm{ref}} \times N_{\mathrm{lon}} \times D}$ as:

$$Q_{\mathrm{nav}} = \mathrm{Projection}(\mathrm{concat}(Q_{\mathrm{lat}}, Q_{\mathrm{lon}})), \tag{3}$$

where $N_{\mathrm{ref}}$ and $N_{\mathrm{lon}}$ denote the number of reference lines and longitudinal anchors, respectively. The navigation query $Q_{\mathrm{nav}}$ and the scene embedding $E_{\mathrm{enc}}$ are then fed into $N$ decoder blocks to model lateral, longitudinal, and cross-modal interactions. Each decoder block is structured as:

$$\begin{aligned} \hat{Q}_{\mathrm{nav}}^{i-1} &= \mathrm{SelfAttn}(\mathrm{SelfAttn}(Q_{\mathrm{nav}}^{i-1}, \dim=0), \dim=1), \\ Q_{\mathrm{nav}}^i &= \mathrm{CrossAttn}(\hat{Q}_{\mathrm{nav}}^{i-1}, E_{\mathrm{enc}}, E_{\mathrm{enc}}). \end{aligned} \tag{4}$$

SelfAttn, CrossAttn denote multi-head self-attention and cross-attention, respectively. Given the decoder's final output $Q_{\mathrm{dec}}$, two MLP heads are applied to produce the CBV-centric multimodal trajectories $\mathcal{T} \in \mathbb{R}^{N_{\mathrm{ref}} \times N_{\mathrm{lon}} \times T \times 6}$ and their confidence scores $\mathcal{S} \in \mathbb{R}^{N_{\mathrm{ref}} \times N_{\mathrm{lon}}}$:

$$\mathcal{T} = \mathrm{MLP}(Q_{\mathrm{dec}}), \ \mathcal{S} = \mathrm{MLP}(Q_{\mathrm{dec}}), \tag{5}$$

where $T$ is the prediction horizon, and each trajectory point $\tau_t^i$ encodes $[p_x, p_y, \cos\theta, \sin\theta, v_x, v_y]$.

## 4 METHODOLOGY

Leveraging the IL pre-trained planning model described in Section 3.2, realistic and multimodal trajectories can be generated across diverse scenarios conditioned on reference lines. However, the open-loop training paradigm leaves the policy vulnerable to covariate shift, even with contrastive learning Halawa et al. (2022); Wang et al. (2023) or data augmentation Cheng et al. (2024a). To address this, we propose *RIFT*, a group-relative RL fine-tuning strategy that enhances style-level controllability and mitigates covariate shift while preserving the trajectory-level realism and route-level controllability from pre-training. The following sections detail *RIFT*'s implementation within the physics-based simulator.

### 4.1 ROUTE-LEVEL INTERACTION ANALYSIS

Following Feng et al. (2023b), we address the "curse of rarity" Liu & Feng (2024) by selectively intervening in a set of critical background vehicles (CBVs) at key moments, while keeping non-critical

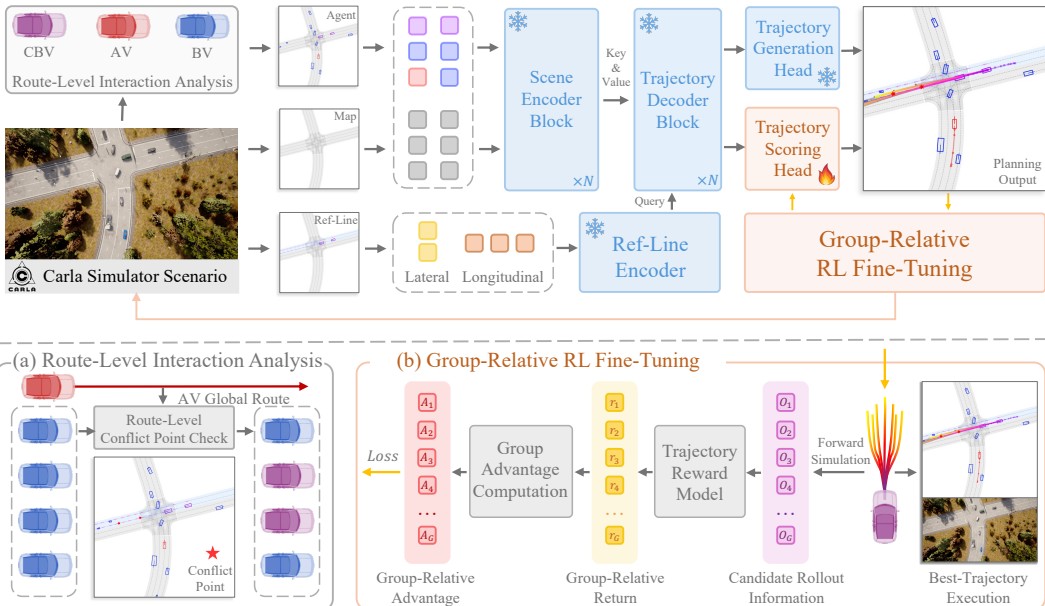

Figure 2: **Overview of the _RIFT_:** Building on the IL pre-trained model, _RIFT_ performs route-level interaction analysis to identify critical background vehicles and the associated reference lines, enabling the generation of realistic and multimodal trajectories. To isolate style-level controllability from the trajectory-level realism and route-level controllability established during pre-training, only the scoring head is fine-tuned via _RIFT_ while freezing other components. Specifically, _RIFT_ computes group-relative advantages over all candidate rollouts, promoting alignment with user-preferred styles and mitigating covariate shift through RL fine-tuning.

agents under rule-based control for efficiency. CBVs are identified via route-level interaction analysis between the AV's predefined global route and the candidate routes of surrounding vehicles, selecting the vehicle with the highest interaction probability (details in Appendix B.2).

The corresponding route-level reference line is then used as a condition for the IL pre-trained planning model (Section 3.2) to synthesize realistic and multimodal trajectories. For each identified CBV, the model generates $N_{\text{ref}} \times N_{\text{lon}}$ candidate trajectories, from which the highest-scoring one is selected for closed-loop execution. This process promotes realistic route-level interactions with the AV and enables the construction of meaningful interactive scenarios.

## 4.2 GROUP-RELATIVE RL FINE-TUNING

Open-loop IL pre-training offers trajectory-level realism and route-level controllability; however, it inevitably suffers from covariate shift in closed-loop deployment, causing error accumulation and unrealistic long-term behaviors. Existing RL Schulman et al. (2017) and hybrid IL–RL methods Peng et al. (2024) partially mitigate covariate shift, but their optimization is restricted to the executed rollout, disregarding alternative candidates and degrading multimodality. More critically, covariate shift induces asymmetric degradation across model components: under the generation–selection paradigm, the generation head, conditioned on route-level priors, remains robust and consistently produces realistic multimodal candidates, whereas the scoring head, trained solely through imitation, is more vulnerable to distribution mismatch. These challenges motivate three key requirements for fine-tuning: (i) preserving multimodality, (ii) addressing asymmetric covariate shift, and (iii) ensuring stable policy improvement. We address these requirements through a unified framework that combines group-relative optimization, asymmetry-aware fine-tuning, and dual-clip stabilization.

To preserve multimodality, we adopt group-relative formulation Shao et al. (2024), which evaluates all candidate modalities within the group and assigns higher relative advantages to those better aligned with user-preferred styles. Considering closed-loop dynamics, we evaluate simulated rollouts rather than raw trajectories to mitigate plan–rollout deviation. Specifically, given $G = N_{\text{ref}} \times N_{\text{lon}}$ candidate trajectories $\mathcal{T} = \{\tau_i\}_{i=1}^{G}$ for a CBV at state $s$, we conduct forward simulation Dauner et al. (2023) (see Appendix B.6) to obtain rollouts $\widetilde{\mathcal{T}} = \{\widetilde{\tau}_i\}_{i=1}^{G}$. Each rollout is evaluated by a user-defined state-wise reward model $\text{StateWiseRM}$, yielding the corresponding discounted returns

$\mathcal{R} = \{R_i\}_{i=1}^G$ from which we derive the group-relative advantages $\mathcal{A} = \{\hat{A}_i\}_{i=1}^G$ as follows:

$$R_i(s) = \sum_{t=0}^T \gamma^t \left[\text{StateWiseRM}(\widetilde{\tau}_i^t, s)\right], \quad \hat{A}_i(s) = \frac{R_i(s) - \text{mean}(\mathcal{R})}{\sqrt{\text{Var}(\mathcal{R}) + \varepsilon}}. \tag{6}$$

Here, $\hat{A}_i$ quantifies the performance of each rollout relative to the group, promoting high-return rollouts without suppressing alternative modes.

In standard GRPO Shao et al. (2024), sampling from the old policy implicitly induces old-policy weighting. Extending this to our enumerated setting involves averaging terms weighted by $\pi_{\theta_{\text{old}}}$ in conjunction with the importance ratio $\rho_i(\theta) = \pi_\theta(\tau_i|s)/\pi_{\theta_{\text{old}}}(\tau_i|s)$, which yields a low-variance estimate of the old-policy expectation over the enumerated support. The aggregated objective is:

$$\mathcal{J}(\theta) = \mathbb{E}_{s\sim\mathcal{D}} \left[ \sum_{i=1}^G \pi_{\theta_{\text{old}}}(\tau_i|s) \min\left[ \rho_i(\theta)\hat{A}_i, \text{clip}\left(\rho_i(\theta), 1-\epsilon, 1+\epsilon\right)\hat{A}_i \right] \right] - \beta\mathbb{D}_{KL}\left[\pi_\theta||\pi_{ref}\right], \tag{7}$$

where $\pi_{\text{ref}}$ denotes the IL pre-trained model. While exact over the enumerated support, this scheme overemphasizes frequent modes and under-represents rare but high-return ones, causing mode collapse and reduced diversity. To balance modality contributions, we adopt an equal-weight objective:

$$\mathcal{J}(\theta) = \mathbb{E}_{s\sim\mathcal{D}} \left[ \frac{1}{G}\sum_{i=1}^G \min\left[ \rho_i(\theta)\hat{A}_i, \text{clip}\left(\rho_i(\theta), 1-\epsilon, 1+\epsilon\right)\hat{A}_i \right] \right] - \beta\mathbb{D}_{KL}\left[\pi_\theta||\pi_{ref}\right]. \tag{8}$$

Under equal weighting, $\rho_i(\theta)$ regulates candidate updates rather than serving as a pure importance weight, removing old-policy bias and yielding balanced updates that preserve multimodality.

To address asymmetric covariate shift, we freeze the generation head to retain trajectory-level realism and fine-tune only the scoring head to enhance style-level controllability. In this setting, constraining the scoring head with the KL term to the IL pre-trained model would anchor learning to a biased reference, thereby hindering adaptation. We therefore remove the KL term, allowing the scoring head to adapt freely while leveraging the stable candidates provided by the frozen generation head.

Removing the KL term improves flexibility but raises stability concerns. Although the clipped-ratio mechanism in PPO constrains update magnitude, it proves insufficient in the group-relative setting. Specifically, when a rare trajectory under the old policy receives a higher probability from the current policy despite a negative advantage, the product $\rho_i(\theta)\hat{A}_i$ can become disproportionately large and destabilize learning. To address this, we incorporate the dual-clip surrogate from Dual-Clip PPO Ye et al. (2020); Gao et al. (2021), which lower-bounds clipped negative advantages. This establishes a trust-region-like constraint that guarantees bounded per-candidate updates (see Theorem A.3), thereby preventing extreme negative shifts while preserving responsiveness to user-preferred styles. The resulting surrogate objective, termed *RIFT*, is

$$\mathcal{J}_{\text{RIFT}}(\theta) = \mathbb{E}_{s\sim\mathcal{D}} \left[ \frac{1}{G}\sum_{i=1}^G \psi\left(\rho_i(\theta), \hat{A}_i\right) \right],$$

$$\psi(\rho, \hat{A}) = \begin{cases} \min\left(\rho\hat{A}, \text{clip}(\rho, 1-\epsilon, 1+\epsilon)\hat{A}\right), & \hat{A} \geq 0, \\ \max\left(\min\left(\rho\hat{A}, \text{clip}(\rho, 1-\epsilon, 1+\epsilon)\hat{A}\right), c\hat{A}\right), & \hat{A} < 0 \end{cases} \quad (\epsilon > 0, \ c > 1). \tag{9}$$

This objective integrates multimodality preservation, asymmetry-aware fine-tuning, and stable optimization into a unified framework, enhancing style-level controllability and mitigating covariate shift while retaining trajectory-level realism and route-level controllability (analysis in Appendix A).

## 5 EXPERIMENT

This section systematically addresses the following research questions: **Q1**: How does *RIFT* compare with representative baselines in terms of the realism and controllability of the generated traffic scenarios? **Q2**: How can the generated traffic scenario be effectively utilized to support downstream autonomous driving tasks? **Q3**: How do the components of *RIFT* contribute to overall performance, and to what extent is style-level controllability preserved under varying user-specified driving styles?

### 5.1 EXPERIMENT SETUPS

Under the dual-stage AV-centric simulation framework, we adopt Pluto Cheng et al. (2024a) as our planning model for its well-established performance and open-source implementation. To ensure

Table 1: **Comparison in Controllability and Realism.** Metrics are evaluated under the PDM-Lite Beißwenger (2024) AV setting across three random seeds, with the **best** and the second-best results highlighted accordingly.

| Method | Type | Kinematic Metrics | | | Interaction Metrics | | | | Map Metrics |
|---|---|---|---|---|---|---|---|---|---|
| | | S-SW ↑ | S-WD ↓ | A-SW ↑ | CPK ↓ | RP ↑ | 2D-TTC ↑ | ACT ↑ | ORR ↓ |
| Pluto | IL | 0.88 ± 0.01 | 5.81 ± 0.06 | 0.90 ± 0.01 | 5.06 ± 2.69 | 564.14 ± 114.41 | 2.50 ± 1.48 | 2.44 ± 1.39 | 0.24 ± 0.15 |
| PPO | RL | 0.95 ± 0.01 | 4.45 ± 0.15 | 0.89 ± 0.02 | 13.95 ± 2.34 | 409.51 ± 30.38 | 2.59 ± 1.60 | 2.52 ± 1.57 | 9.17 ± 2.39 |
| FREA | RL | 0.93 ± 0.01 | 5.10 ± 0.14 | 0.93 ± 0.01 | 30.42 ± 5.28 | 292.81 ± 68.54 | 2.71 ± 1.40 | 2.67 ± 1.41 | 9.01 ± 2.09 |
| FPPO-RS | RL | 0.87 ± 0.01 | 5.80 ± 0.11 | 0.80 ± 0.03 | 21.39 ± 3.23 | 356.79 ± 26.19 | 2.55 ± 1.69 | 2.53 ± 1.68 | 8.60 ± 0.25 |
| SFT-Pluto | SFT | 0.88 ± 0.02 | 6.01 ± 0.19 | 0.87 ± 0.02 | 6.33 ± 2.23 | 780.48 ± 41.05 | 2.20 ± 1.64 | 2.12 ± 1.51 | 0.06 ± 0.07 |
| RS-Pluto | SFT+RLFT | 0.93 ± 0.00 | 5.40 ± 0.15 | 0.92 ± 0.01 | 4.11 ± 3.90 | 819.40 ± 74.07 | 2.27 ± 1.45 | 2.23 ± 1.43 | 1.05 ± 0.31 |
| RTR-Pluto | SFT+RLFT | 0.85 ± 0.00 | 6.24 ± 0.16 | 0.81 ± 0.03 | 6.98 ± 2.59 | 481.60 ± 70.19 | 2.55 ± 1.60 | 2.47 ± 1.51 | 0.08 ± 0.09 |
| PPO-Pluto | RLFT | 0.95 ± 0.01 | 4.96 ± 0.31 | 0.90 ± 0.02 | 6.89 ± 3.19 | 683.57 ± 38.12 | 2.66 ± 1.50 | 2.60 ± 1.43 | 0.07 ± 0.13 |
| REINFORCE-Pluto | RLFT | 0.92 ± 0.01 | 5.63 ± 0.19 | 0.90 ± 0.02 | 6.98 ± 0.86 | 813.70 ± 24.76 | 2.39 ± 1.64 | 2.30 ± 1.55 | 1.37 ± 1.13 |
| GRPO-Pluto | RLFT | 0.94 ± 0.04 | 4.96 ± 0.89 | 0.96 ± 0.00 | 7.24 ± 4.04 | 892.65 ± 65.27 | 2.65 ± 1.44 | 2.61 ± 1.48 | 0.10 ± 0.08 |
| RIFT-Pluto (ours) | RLFT | 0.97 ± 0.01 | 4.46 ± 0.43 | 0.93 ± 0.01 | 6.83 ± 2.62 | 995.33 ± 84.62 | 2.74 ± 1.30 | 2.71 ± 1.32 | 0.36 ± 0.20 |

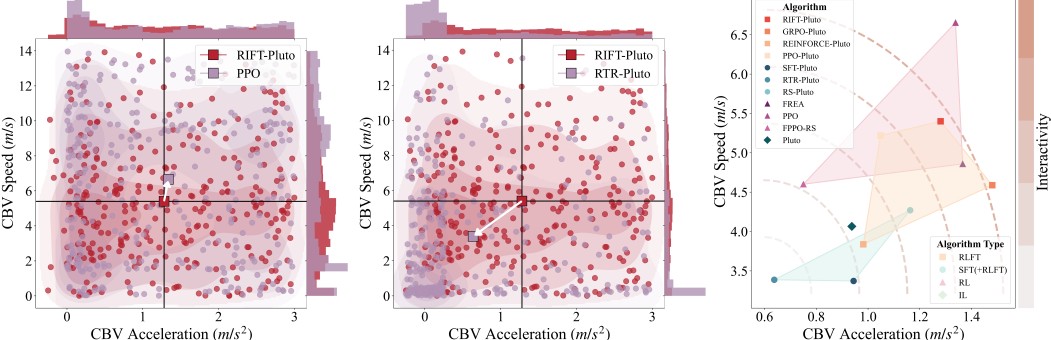

Figure 3: **Speed and Acceleration Distribution.** RL-based methods tend to be interactive but unnatural, whereas supervised methods are overly conservative. *RIFT* strikes a balance, yielding higher interactivity with realistic distributional profiles, reducing hesitation while maintaining safe interactions.

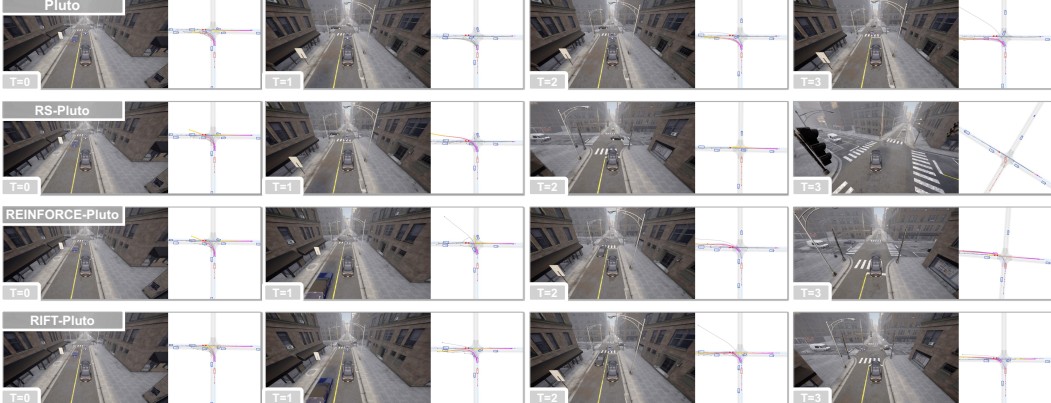

Figure 4: Temporal comparisons illustrating *RIFT*'s superior performance over other baselines under AV-centric closed-loop simulation. CBV is marked in purple, AV (PDM-Lite) is in red, and BVs are in blue.

fair comparison, we use the official IL pre-trained checkpoint provided by Pluto, trained on the nuPlan dataset Caesar et al. (2021). Simulations are conducted in CARLA Dosovitskiy et al. (2017), leveraging Bench2Drive Jia et al. (2024) to support AV-centric closed-loop simulation and evaluation. Implementation details, training protocols, and evaluation settings are described in Appendix B.

**Baseline.** To systematically evaluate the effectiveness of *RIFT* in traffic simulation, we compare it against the following baselines, with implementation details provided in Appendix B.5.

- **Pure RL/IL**: Methods trained solely with RL or IL, without fine-tuning, including *Pluto* Cheng et al. (2024a), as well as *FREA*, *FPPO-RS*, and *PPO*, all from Chen et al. (2024b).

- **RLFT/SFT**: Methods that fine-tune the pre-trained Pluto model using either RL or supervised objectives, including *PPO-Pluto* Schulman et al. (2017), *REINFORCE-Pluto* Sutton et al. (1999), *GRPO-Pluto* Shao et al. (2024), and *SFT-Pluto*.
- **Hybrid**: Methods that combine RL and supervised fine-tuning, including *RTR-Pluto* Zhang et al. (2023a) and *RS-Pluto* Peng et al. (2024).

All methods are fine-tuned on the scoring head to ensure fair comparisons, while isolating style-level controllability from trajectory-level realism and route-level controllability, as confirmed by the ablation studies in Section 5.4. Following the realism standards of the Sim Agent Challenge in WOSAC Montali et al. (2023), we adopt a normal style reward for all RL-based baselines, with details in Appendix B.7. Results under an aggressive style reward are reported in Section 5.4.

**Metrics**. Building on the WOSAC evaluation framework, we categorize our evaluation metrics into three groups: *kinematic metrics*, *interaction metrics*, and *map metrics*. Kinematic metrics capture distributional motion properties (S-SW, S-WD, A-SW), as in Chen et al. (2024a), with the absence of ground-truth trajectories in CARLA precluding displacement-based measures (e.g., ADE, FDE). Interaction metrics evaluate agent interactions through collision frequency (Collision Per Kilometer, CPK), driving efficiency (Route Progress, RP), and safety-critical measures, including 2D-TTC Guo et al. (2023) and ACT Venthuruthiyil & Chunchu (2022). Map metrics evaluate adherence to road geometry through the Off-Road Rate (ORR). Collectively, these metrics comprehensively evaluate realism and controllability in closed-loop simulation; detailed definitions are in Appendix B.8.

## 5.2 Realistic and Controllable Traffic Scenario Generation (Q1)

**Main Results.** To address **Q1**, we evaluate the controllability and realism of the generated scenario across CBV methods, with results summarized in Table 1. *RIFT* consistently outperforms all baselines in both aspects across most settings. While supervised learning methods achieve slightly lower CPK and ORR, this improvement is primarily due to their inherently conservative behavior, derived from the expert PDM-Lite Beißwenger (2024), which prioritizes safety by avoiding risky maneuvers.

This conservative tendency is further highlighted in Figure 3, where supervised policies exhibit significantly lower speed and acceleration profiles. In contrast, *RIFT* strikes a more favorable balance between safety and interactivity. It achieves superior safety performance, as reflected by higher 2D-TTC and ACT scores, while avoiding the overly cautious behaviors typical of supervised approaches. As shown in Figure 3, *RIFT* demonstrates higher average speed and acceleration, indicating more interactive behavior, while maintaining realistic motion profiles.

**Qualitative Results.** To further demonstrate the effectiveness of *RIFT*, we compare closed-loop simulations against representative baselines, as shown in Figure 4. Baseline methods often suffer from unstable or low-quality trajectory selection in closed-loop settings, whereas *RIFT* consistently selects smooth, high-quality trajectories with superior temporal consistency. Further qualitative examples are presented in Appendix D.5.

**Covariate Shift Analysis.** a key challenge for reliable long-term traffic simulation is the open-loop vs. closed-loop covariate shift that arises when IL pre-trained policies are deployed for long-horizon rollouts. As shown in Figure 3 and Figure 4, the IL pre-trained Pluto exhibits abnormally low speeds, early braking, and difficulty re-accelerating—clear symptoms of mismatch between open-loop training distributions and closed-loop execution.

Supervised fine-tuning partially alleviates these issues but remains inherently conservative due to the safety-oriented PDM-Lite expert, consistent with the lower CPK and ORR observed in Table 1. In contrast, *RIFT* directly optimizes under closed-loop rollouts, reducing this distributional mismatch and achieving higher speed and acceleration while maintaining realistic motion. These findings indicate that RL-based fine-tuning more effectively corrects covariate shift than supervised approaches.

## 5.3 Generated Traffic Scenarios for Closed-Loop AV Evaluation (Q2)

To address **Q2**, we assess the suitability of traffic scenarios generated by different CBV methods for closed-loop AV evaluation. Following KING Hanselmann et al. (2022), we adopt PDM-Lite Beißwenger (2024)—a rule-based planner with privileged access—as a reference to evaluate

Table 2: **Comparison of AV Evaluation across CBV Methods.** Each metric is evaluated across three random seeds, with the **best** and the second-best results highlighted accordingly.

| Method | PDM-Lite | | PlanT | | UniAD | | VAD | |
|---|---|---|---|---|---|---|---|---|
| | DS ↑ | BR ↓ | DS | ΔDS ↓ | DS | ΔDS ↓ | DS | ΔDS ↓ |
| Pluto | 77.84 ± 2.20 | 23.33 ± 5.77 | 42.52 ± 4.72 | -35.32 | 73.73 ± 1.24 | -4.11 | 66.87 ± 2.11 | -10.97 |
| PPO | 76.26 ± 0.12 | 30.00 ± 0.00 | 36.39 ± 1.11 | -39.87 | 69.79 ± 1.41 | -6.47 | 67.64 ± 1.27 | -8.62 |
| FREA | 83.53 ± 0.13 | 20.00 ± 0.00 | 39.61 ± 1.34 | -43.92 | 69.29 ± 5.22 | **-14.24** | 67.57 ± 5.37 | -15.96 |
| FPPO-RS | 83.52 ± 0.09 | 20.00 ± 0.00 | 38.85 ± 4.91 | -44.67 | 75.13 ± 5.18 | -8.39 | 69.15 ± 2.79 | -14.37 |
| SFT-Pluto | 86.09 ± 2.04 | 13.33 ± 5.77 | 39.41 ± 4.97 | -47.28 | 77.49 ± 5.93 | -9.20 | 68.89 ± 0.87 | -17.80 |
| RS-Pluto | 89.32 ± 1.41 | 13.33 ± 5.77 | 42.05 ± 4.08 | -47.27 | 80.62 ± 0.78 | -8.70 | 69.48 ± 5.02 | -19.84 |
| RTR-Pluto | 87.64 ± 1.56 | 10.00 ± 0.00 | 40.08 ± 2.38 | **-47.56** | 77.69 ± 2.82 | -9.95 | 66.27 ± 4.53 | -21.37 |
| PPO-Pluto | 85.63 ± 2.02 | 16.67 ± 5.77 | 41.86 ± 2.78 | -43.77 | 77.14 ± 3.36 | -8.49 | 68.62 ± 3.16 | -17.01 |
| REINFORCE-Pluto | **92.17** ± 3.45 | 10.00 ± 10.00 | 45.25 ± 1.75 | -46.92 | 79.89 ± 1.97 | -12.28 | 70.28 ± 3.58 | **-21.89** |
| GRPO-Pluto | 89.86 ± 2.10 | **6.67** ± 5.77 | 47.24 ± 5.67 | -42.62 | 81.02 ± 0.64 | -8.84 | 72.55 ± 0.74 | -17.31 |
| RIFT-Pluto (ours) | **94.78** ± 1.37 | **0.00** ± 0.00 | 44.28 ± 3.15 | **-50.50** | 73.79 ± 6.53 | **-20.99** | 68.24 ± 3.23 | **-26.54** |

Table 3: **Ablation Study on *RIFT*.** Evaluation under PDM-Lite AV setting with three random seeds.

| Method | Kinematic Metrics | | | Interaction Metrics | | | | Map Metrics |
|---|---|---|---|---|---|---|---|---|
| | S-SW ↑ | S-WD ↓ | A-SW ↑ | CPK ↓ | RP ↑ | 2D-TTC ↑ | ACT ↑ | ORR ↓ |
| w/ Old-Weight | 0.82 (-0.15) | 6.24 (+1.78) | 0.85 (-0.08) | 7.51 (+0.68) | 574.51 (-420.82) | 2.70 (-0.04) | 2.68 (-0.03) | 0.00 (-0.36) |
| w/ All-Head | 0.96 (-0.01) | 4.70 (+0.24) | 0.94 (+0.01) | 7.84 (+1.01) | 827.12 (-168.21) | 2.83 (+0.09) | 2.76 (+0.05) | 0.43 (+0.07) |
| w/ KL | 0.93 (-0.04) | 5.33 (+0.87) | 0.90 (-0.03) | 7.05 (+0.22) | 815.06 (-180.27) | 2.76 (+0.02) | 2.73 (+0.02) | 0.38 (+0.02) |
| w/ PPO-Clip | 0.91 (-0.06) | 5.92 (+1.46) | 0.94 (+0.01) | 2.03 (-4.80) | 655.39 (-339.94) | 2.57 (-0.17) | 2.54 (-0.17) | 0.04 (-0.32) |
| w/ Aggressive | 0.97 (+0.00) | 3.89 (-0.57) | 0.94 (+0.01) | 8.41 (+1.58) | 1053.76 (+58.43) | 2.93 (+0.19) | 2.88 (+0.17) | 0.91 (+0.55) |
| RIFT-Pluto (ours) | 0.97 | 4.46 | 0.93 | 6.83 | 995.33 | 2.74 | 2.71 | 0.36 |

two key scenario properties: feasibility, measured by Driving Score (DS), and naturalness, captured by Blocked Rate (BR). A high DS indicates that the AV can reliably complete the scenario, while a low BR reflects realistic interactions without excessive obstruction from surrounding vehicles. **Importantly, only the BR measured under a reliable rule-based planner isolates the naturalness of the traffic itself, as such a planner does not introduce self-induced stalls.** Together, DS and BR offer a principled basis for evaluating scenario quality.

To further assess the ability of each scenario to reveal weaknesses in learning-based planners, we compare PlanT Renz et al. (2022), UniAD Hu et al. (2023), and VAD Jiang et al. (2023a) with PDM-Lite. As these models are sensitive to subtle or adversarial interactions, informative scenarios should induce noticeable performance drops. As shown in Table 2, traffic generated by *RIFT* achieves the highest DS and lowest BR under PDM-Lite, while also causing the largest degradation across all learning-based planners. These results confirm that *RIFT* generates interactive and feasible scenarios that effectively expose limitations of modern AV systems. See Appendix C for detailed results.

## 5.4 ABLATION STUDY (Q3)

Building on the design choices introduced in Section 4.2, we systematically ablate five components of *RIFT*: weighting scheme (Old-Weight vs. Equal-Weight), fine-tuning module (Scoring Head vs. All Head), KL regularization (w/ KL vs. w/o KL), policy clipping (Dual-Clip vs. PPO-Clip), and style preference (Normal vs. Aggressive). All experiments share identical settings, and results are reported in Table 3.

**Equal-Weight *vs.* Old-Weight.** Replacing old-policy weighting with equal weighting eliminates the likelihood bias toward frequent modes and enables balanced updates across all candidates. This leads to improved exploitation of high-return rollouts and better multimodality preservation.

**Scoring Head *vs.* All Head.** Freezing the generation head is crucial for retaining trajectory-level realism and route-level controllability. Fine-tuning all heads (*w/ All Head*) disrupts the pre-trained generation head and slightly degrades realism metrics, whereas fine-tuning only the scoring head achieves better controllability without compromising realism.

**w/ KL *vs.* w/o KL.** Anchoring the scoring head to the IL pre-trained reference via KL regularization (*w/ KL*) constrains adaptation to a biased reference under asymmetric covariate shift. Removing

this term improves controllability while maintaining realism, confirming that free adaptation of the scoring head yields more effective policy improvement.

**Dual-Clip** *vs.* **PPO-Clip.** Replacing dual-clip with standard PPO clipping (*w/ PPO-Clip*) results in overly conservative behaviors and reduced efficiency, as extreme negative updates can dominate and suppress positive learning signals. Dual-clip bounds such updates while preserving responsiveness to high-return rollouts, producing more realistic and efficient behavior.

**Normal** *vs.* **Aggressive.** Adopting a more aggressive reward that emphasizes efficiency increases route progress but also raises collision and off-road rates, illustrating the efficiency–safety trade-off. These results demonstrate that *RIFT* supports flexible style shaping while maintaining stability and multimodality. Additional qualitative insights on controllability are provided in Appendix D.1.

## 6 CONCLUSION

In this work, we propose a dual-stage AV-centric simulation framework that conducts IL pre-training in a data-driven simulator to capture trajectory-level realism and route-level controllability, followed by RL fine-tuning in a physics-based simulator to address covariate shift and enhance style-level controllability. During fine-tuning, we introduce *RIFT*, a novel group-relative RL fine-tuning strategy that evaluates all candidate modalities using the group-relative formulation combined with a surrogate objective for optimization, thereby enhancing style-level controllability and mitigating covariate shift, while preserving the trajectory-level realism and route-level controllability established in IL pre-training. Extensive experiments demonstrate that *RIFT* generates scenarios with superior realism and controllability, effectively revealing the limitations of modern AV systems and further bridging the gap between traffic simulation and reliable closed-loop evaluation. Due to space limits, limitations and future directions are in Appendix E.1, and experimental reproducibility details are in Appendix B.

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

## A  THEORETICAL ANALYSIS

### A.1  SETTING

For each $s \sim \mathcal{D}$, a frozen trajectory generation head yields $\mathcal{C}(s) = \{\tau_i\}_{i=1}^G$. The trajectory score head defines $\pi_\theta(\tau_i \mid s)$ on $\mathcal{C}(s)$. Finite-horizon simulation provides returns

$$R_i(s) = \sum_{t=0}^{T} \gamma^t \, \text{StateWiseRM}(\widetilde{\tau}_i^t, s). \tag{10}$$

Uniform (within-group) moments:

$$\mu_{\text{uni}}(s) = \tfrac{1}{G} \sum_{j=1}^G R_j(s), \qquad \sigma_{\text{uni}}^2(s) = \tfrac{1}{G} \sum_{j=1}^G (R_j(s) - \mu_{\text{uni}}(s))^2. \tag{11}$$

Uniform, centered advantages:

$$\hat{A}_i(s) = \frac{R_i(s) - \mu_{\text{uni}}(s)}{\sqrt{\sigma_{\text{uni}}^2(s) + \varepsilon}}, \quad \frac{1}{G} \sum_{i=1}^G \hat{A}_i(s) = 0. \tag{12}$$

Let $\rho_i(\theta) = \pi_\theta(\tau_i \mid s) / \pi_{\theta_{\text{old}}}(\tau_i \mid s)$. Define the *RIFT* surrogate

$$\mathcal{J}_{\text{RIFT}}(\theta) = \mathbb{E}_s \left[ \frac{1}{G} \sum_{i=1}^G \psi\big(\rho_i(\theta), \hat{A}_i(s)\big) \right], \tag{13}$$

with dual-clip kernel

$$\psi(\rho, \hat{A}) = \begin{cases} \min\big(\rho \hat{A}, \, \text{clip}(\rho, 1 - \epsilon, 1 + \epsilon) \, \hat{A}\big), & \hat{A} \geq 0, \\ \max\big( \min\big(\rho \hat{A}, \, \text{clip}(\rho, 1 - \epsilon, 1 + \epsilon) \, \hat{A}\big), \, c \, \hat{A}\big), & \hat{A} < 0, \end{cases} \quad (\epsilon > 0, \; c > 1). \tag{14}$$

**Assumptions.**  (A) Support floor: there exists $\pi_{\min} > 0$ such that $\pi_{\theta_{\text{old}}}(\tau_i \mid s) \geq \pi_{\min}$ for all $(s, i)$, and $\pi_\theta > 0 \Rightarrow \pi_{\theta_{\text{old}}} > 0$ on $\mathcal{C}(s)$.  (B) Boundedness: $|\hat{A}_i(s)| \leq A_{\max}$.  (C) Regularity: $\log \pi_\theta(\tau_i \mid s)$ is $L$-Lipschitz and $C^2$ on compact $\Theta$.

### A.2  LISTWISE VIEW AND DIVERSITY PRESSURE

Consider the unclipped uniform surrogate

$$L_{\text{RIFT}}(\theta) = \mathbb{E}_s \left[ \frac{1}{G} \sum_{i=1}^G \rho_i(\theta) \, \hat{A}_i(s) \right] = \mathbb{E}_s \left[ \frac{1}{G} \sum_{i=1}^G \frac{\hat{A}_i(s)}{\pi_{\theta_{\text{old}}}(\tau_i \mid s)} \pi_\theta(\tau_i \mid s) \right]. \tag{15}$$

**Proposition A.1** (Pairwise ascent and diversity).  *Fix $s$ and shift an infinitesimal mass $\delta$ from $j$ to $i$ in $\pi_\theta(\cdot \mid s)$. Then $\delta L_{\text{RIFT}}(\theta) = \frac{\delta}{G} \left( \frac{\hat{A}_i(s)}{\pi_{\theta_{\text{old}}}(\tau_i | s)} - \frac{\hat{A}_j(s)}{\pi_{\theta_{\text{old}}}(\tau_j | s)} \right)$. Hence ascent moves mass toward larger $\hat{A}/\pi_{\text{old}}$, amplifying underrepresented high-quality candidates when $\pi_{\text{old}}$ is peaky.*

**Corollary A.2** (Top-1 Fisher consistency under uniform reference).  *If $\pi_{\theta_{\text{old}}}$ is uniform on $\mathcal{C}(s)$ and $i^\star(s) = \arg\max_i \hat{A}_i(s)$ is unique, any global maximizer of $L_{\text{RIFT}}$ concentrates $\pi_\theta(\cdot \mid s)$ on $i^\star(s)$.*

### A.3  CLIPPING AS STABILITY CONTROL

Clipping is a pointwise pessimistic transform: for any $x = \rho \hat{A}$,

$$\min(\rho \hat{A}, \text{clip}(\rho) \hat{A}) \leq \rho \hat{A}.$$

Summed over mixed signs, there is no global monotone lower bound for $L_{\text{RIFT}}$; instead, clipping serves to bound the value and the gradient.

**Lemma A.3** (Bounded values and gradients).  *If $|\hat{A}| \leq A_{\max}$, then for all $(s, i)$: (i) Value bounds: $\psi \in [0, (1 + \epsilon)\hat{A}]$ for $\hat{A} \geq 0$, and $\psi \in [c \hat{A}, 0]$ for $\hat{A} < 0$. (ii) Gradient bounds:*

$$\left| \frac{\partial \psi}{\partial \log \pi_\theta} \right| \leq \begin{cases} (1 + \epsilon)|\hat{A}|, & \hat{A} \geq 0, \\ c \, |\hat{A}|, & \hat{A} < 0, \end{cases}$$

*and on the negative branch when the dual-clip is active ($\psi = c \hat{A}$) the derivative is $0$.*

## A.4 SMOOTHNESS W.R.T. POLICY DIVERGENCE

Write $w_i(s) = \hat{A}_i(s)/\pi_{\theta_{\text{old}}}(\tau_i \mid s)$ and note $|w_i| \leq A_{\max}/\pi_{\min}$ under Assumption A with $\pi_{\min} = \inf_{s,i} \pi_{\theta_{\text{old}}}(i \mid s) > 0$ (label-smoothing in practice). Then the unclipped surrogate is linear in $\pi_\theta$:

$$L_{\text{RIFT}}(\theta) - L_{\text{RIFT}}(\theta') = \mathbb{E}_s\left[\frac{1}{G}\sum_i w_i(s)\big(\pi_\theta(i \mid s) - \pi_{\theta'}(i \mid s)\big)\right].$$

**Lemma A.4** (Lipschitz continuity via KL). *For any $\theta, \theta'$,*

$$\big|L_{\text{RIFT}}(\theta) - L_{\text{RIFT}}(\theta')\big| \leq \frac{A_{\max}}{\pi_{\min}}\sqrt{2\,\mathbb{E}_s\big[\text{KL}\big(\pi_\theta(\cdot \mid s)\,\|\,\pi_{\theta'}(\cdot \mid s)\big)\big]}.$$

*Proof.* By Hölder and Pinsker: $|\sum_i w_i \Delta\pi| \leq \|w\|_\infty \|\Delta\pi\|_1 \leq (A_{\max}/\pi_{\min})\sqrt{2\,\text{KL}(\pi_\theta\|\pi_{\theta'})}$, then average over $s$. □

**Lemma A.5** (Lipschitz continuity of clipped surrogate). *Because $\partial\psi/\partial\pi_\theta(i|s)$ is bounded by $A_{\max}/\pi_{\min}$ whenever the active branch is differentiable,*

$$\big|\mathcal{J}_{\text{RIFT}}(\theta) - \mathcal{J}_{\text{RIFT}}(\theta')\big| \leq \frac{A_{\max}}{\pi_{\min}}\sqrt{2\,\mathbb{E}_s\big[\text{KL}\big(\pi_\theta(\cdot \mid s)\,\|\,\pi_{\theta'}(\cdot \mid s)\big)\big]}.$$

## A.5 VARIANCE AND ENUMERATION

Let $f_i(s; \theta) = \psi(\rho_i(\theta), \hat{A}_i(s))$. Exact enumeration yields $\text{Var}\big(\frac{1}{G}\sum_i f_i \mid s\big) = 0$ (assuming $\widetilde{\tau}_i$ and their evaluations are fixed during the update; otherwise, environment randomness still induces nonzero variance), while sampling $i$ i.i.d. within the group gives conditional variance $\text{Var}(f_i \mid s)/N$ for $N$ samples.

## A.6 CONVERGENCE OF STOCHASTIC ASCENT

**Theorem A.6** (Convergence to a stationary point). *Under Assumptions A–C, with step sizes $\eta_k > 0$, $\sum_k \eta_k = \infty$, $\sum_k \eta_k^2 < \infty$, and unbiased bounded-variance stochastic subgradients, the iterates of stochastic subgradient ascent on $\mathcal{J}_{\text{RIFT}}$ satisfy*

$$\liminf_{k\to\infty} \mathbb{E}\big[\text{dist}\big(0, \partial^C \mathcal{J}_{\text{RIFT}}(\theta_k)\big)\big] = 0,$$

*where $\partial^C$ denotes the Clarke generalized gradient.*

*Sketch.* By Lemma A.3, generalized gradients are uniformly bounded; regularity of $\log \pi_\theta$ on compact $\Theta$ implies Lipschitz continuity. Robbins–Monro / Kushner–Yin results for non-smooth stochastic approximation apply. □

## A.7 WHY RIFT PRESERVES MULTIMODALITY

By Proposition A.1, ascent compares $\hat{A}/\pi_{\text{old}}$: under peaky $\pi_{\text{old}}$, underrepresented high-$\hat{A}$ candidates receive stronger positive updates, preserving and enhancing diversity. In the special case $\pi_{\text{old}}$ is (approximately) uniform, *RIFT* reduces to a listwise ranking ascent that directly promotes larger $\hat{A}$.

# B EXPERIMENTAL DETAILS

## B.1 EXPERIMENT FRAMEWORK

Our framework for reliable AV-centric closed-loop simulation is developed upon well-established traffic simulation platforms, notably the CARLA Leaderboard Team (2025) and Bench2Drive Jia et al. (2024), which serve as standard benchmarks in autonomous driving research. Traditionally, these platforms use predefined scenarios along the AV's global route to evaluate the multi-dimensional performance of AV methods. In contrast, we replace these static scenarios with dynamically generated traffic flows by randomly spawning background vehicles around the AV's global path and simulating

their behavior using rule-based driving policies, as described in Section 3.1. Through the CBV identification mechanism outlined in Appendix B.2, we naturally introduce interactions between the AV and CBVs, thereby generating continuous, interactive scenarios over time. This framework serves as the foundation for both the training and evaluation processes in this paper.

## B.2 Route-level Analysis for CBV Identification

Identifying Critical Background Vehicles (CBVs) is essential to our AV-centric closed-loop simulation. Let $\mathcal{V}_{\text{AV}}$ denote the autonomous vehicle (AV), and $\mathcal{V}_{\text{BV}} = \{\mathcal{V}_i\}_{i=1}^N$ represent the set of background vehicles in the environment. The AV navigates along a predefined global route $\mathcal{P} = \{p_k\}_{k=1}^M$, where each $p_k$ corresponds to a waypoint along the route. The goal of CBV identification is to select background vehicles that are likely to share the AV's destination and have similar estimated travel distance, thereby facilitating route-level interactions between the AV and CBVs. The primary criterion for identifying CBVs is the relative *distance-to-goal* difference between the AV and each background vehicle. This is mathematically expressed as:

$$\left| \hat{D}_{\text{global}}(p_k, \mathcal{V}_i) - \hat{D}_{\text{global}}(p_k, \mathcal{V}_{\text{AV}}) \right| < \delta, \tag{16}$$

where, $\hat{D}_{\text{global}}(p_k, \mathcal{V}_i)$ and $\hat{D}_{\text{global}}(p_k, \mathcal{V}_{\text{AV}})$ denote the estimated travel distance required for the background vehicle $\mathcal{V}_i$ and the AV to reach waypoint $p_k$, respectively. The distance-to-goal for each vehicle is computed by determining the distance from its current position to the target waypoint $p_k$ using the A* global path planning algorithm Hart et al. (1968). A threshold $\delta$ is introduced to define the maximum allowable difference in distance-to-goal. A background vehicle is considered critical and included in the CBV set $\mathcal{C}$ if the absolute distance-to-goal difference between it and the AV is smaller than $\delta$.

This approach selects background vehicles whose destinations and estimated travel distances are sufficiently aligned with those of the AV, thereby ensuring meaningful and realistic route-level interactions. Once a CBV is identified, the planning path previously generated via A* during distance-to-goal estimation is directly adopted as its global navigation path, which is further transformed into the reference line for downstream CBV planning, naturally introducing route-level interactions between the AV and CBVs. The threshold $\delta$ serves as a tunable parameter to adjust the sensitivity of the CBV selection process. In this study, we set $\delta$ to $15m$ to achieve a balanced trade-off between sensitivity and selection accuracy.

**Limitations and Future Directions.** A key limitation of our current CBV identification module is its reliance on route-level overlap when selecting interacting vehicles. While this rule provides stable and semantically interpretable interaction contexts, it is inherently conservative: in complex intersections, cross-traffic vehicles whose routes do not geometrically overlap with the ego's route may still come into close proximity and exert strong interaction influence, yet remain excluded under this criterion. This highlights the need for more comprehensive interaction-mining strategies that incorporate dynamic proximity, safety-critical cues, or conflict-based reasoning to capture non-overlapping but behaviorally significant agents. Because CBV identification in our framework is fully decoupled from route-conditioned CBV planning, such enhanced, intersection-aware mechanisms can be integrated seamlessly in future work without altering the core learning pipeline. Promising future directions include leveraging VLM-assisted semantic risk analysis to identify behaviorally significant agents and infer their corresponding interaction routes in a more context-aware manner.

## B.3 Algorithm Framework

For clarity, we summarize the procedure of *RIFT* within our AV-centric closed-loop simulation framework in Algorithm 1. The planning model is initialized from the IL pre-trained checkpoint provided by Pluto official codebase[1], followed by RL fine-tuning within the CARLA simulator Dosovitskiy et al. (2017) to generate realistic and controllable traffic scenarios.

---

[1] https://github.com/jchengai/pluto

---

**Algorithm 1** Procedure for *RIFT* in the AV-Centric Closed-Loop Simulation Framework.

---

1: **Input**: IL pre-trained planning model $\pi_{\theta_{\text{init}}}$, buffer $\mathcal{D}$  $\triangleright$ IL pre-training (nuPlan Caesar et al. (2021))
2: planning model $\pi_\theta \leftarrow \pi_{\theta_{\text{init}}}$
3: **for** iteration $= 1, \ldots, I$ **do**  $\triangleright$ RL fine-tuning (CARLA Dosovitskiy et al. (2017))
4:     Update the old planning model $\pi_{\theta_{old}} \leftarrow \pi_\theta$
5:     **while** $\mathcal{D}$ not full **do**  $\triangleright$ Collect rollout data
6:         **for** step $= 1, \ldots, T$ **do**
7:             Obtain $G$ candidate trajectories $\{\tau_i\}_{i=1}^G$ from $\pi_{\theta_{old}}$ for each CBV $\triangleright$ Policy inference
8:             Compute simulated rollouts $\{\widetilde{\tau}_i\}_{i=1}^G$ from $\{\tau_i\}_{i=1}^G$  $\triangleright$ Forward simulation
9:             Compute reward $\{R_i\}_{i=1}^G$, advantage $\{\hat{A}_i\}_{i=1}^G$ for each $\widetilde{\tau}_i$ with Equation (6)
10:            Store transition into buffer $\mathcal{D}$
11:         **end for**
12:     **end while**
13:     **for** *RIFT* iteration $= 1, \ldots, \mu$ **do**  $\triangleright$ Policy fine-tuning
14:         Sample mini-batches transition from the buffer $\mathcal{D}$
15:         Update model $\pi_\theta$ by maximizing the *RIFT* objective (Equation (9))
16:     **end for**
17: **end for**
18: **Output**: RL fine-tuned planning model

---

### B.4 TRAINING DETAILS

We perform RL fine-tuning on selected modules of the IL pre-trained planning model (Pluto). As shown in the ablation results (Section 5.4), fine-tuning only the trajectory scoring head achieves the best trade-off between realism and controllability. Accordingly, all fine-tuning baselines adopt this setting to ensure consistency and fair comparison. Our training framework is built on the open-source Lightning platform[2]. Fine-tuning is conducted on $2\times$ `Bench2Drive220`, while evaluation is performed on `dev10`, both from the Bench2Drive project. All experiments are conducted on NVIDIA GeForce RTX 4090D GPUs, with each fine-tuning run taking approximately 8 hours on a single GPU. Detailed training setups and hyperparameter configurations are provided in Table 4 and Table 5.

### B.5 BASELINES DETAILED DESCRIPTION

To comprehensively evaluate *RIFT* in an AV-centric closed-loop simulation environment, we compare it against a range of baselines, including pure imitation learning (IL), pure reinforcement learning (RL), and various fine-tuning approaches based on IL, RL, or their combination. We initialize all fine-tuning methods from the pre-trained Pluto checkpoint and fine-tune only the trajectory scoring head to preserve trajectory-level realism. The details of each baseline are summarized below.

- *Pluto* Cheng et al. (2024a) is an open-source IL-based planning framework for autonomous driving. It processes vectorized scene representations as input and outputs multimodal trajectories for downstream planning. In the AV-centric closed-loop simulation, the method directly uses a pre-trained checkpoint without additional fine-tuning.

- *FREA* Chen et al. (2024b) is an RL-based approach designed to generate safety-critical yet AV-feasible scenarios. It incorporates a feasibility-aware training objective. In the AV-centric closed-loop simulation, FREA selects potential collision points along the AV's global route as adversarial goals.

- *PPO* Chen et al. (2024b) is a variant of FREA that focuses solely on generating safety-critical scenarios. Unlike FREA, it disregards the feasibility constraints of AV and treats adversariality as the only optimization objective.

- *FPPO-RS* Chen et al. (2024b) is another FREA variant that integrates AV's feasibility constraints into the reward shaping process, thereby balancing adversariality with scenario reasonableness.

---

[2]https://github.com/Lightning-AI/pytorch-lightning

- *PPO-Pluto* fine-tunes the pre-trained planning model using the PPO algorithm Schulman et al. (2017). The fine-tuning follows the same reward structure as detailed in Appendix B.7, aligning with *RIFT*.

- *REINFORCE-Pluto* employs the REINFORCE algorithm Sutton et al. (1999) to fine-tune the pre-trained Pluto model under the same reward design as detailed in Appendix B.7.

- *GRPO-Pluto* utilizes the basic GRPO algorithm Shao et al. (2024) for fine-tuning, employing the pre-trained Pluto model as the reference for KL regularization, while incorporating the standard PPO-Clip.

- *SFT-Pluto* is a purely supervised fine-tuning approach, where PDM-Lite Beißwenger (2024) serves as the expert model, providing supervision at the target speed level.

- *RTR-Pluto* Zhang et al. (2023a) is a hybrid framework combining imitation and reinforcement learning. While the original RTR utilizes human driving trajectories as supervision, our setting replaces this with PDM-Lite due to the lack of human-level demonstrations. The RL component uses sparse infraction-based rewards, consistent with the original RTR, and applies PPO for optimization.

- *RS-Pluto* Peng et al. (2024) also adopts a hybrid IL+RL paradigm, originally trained via RE-INFORCE using ground-truth supervision and sparse rewards to ensure safety and realism. In our adaptation, PDM-Lite substitutes the ground-truth expert, while the rest of the methodology remains unchanged.

## B.6 FORWARD SIMULATION

Trajectory-based imitation learning often overlooks underlying system dynamics, leading to discrepancies between planned and executed behavior Cheng et al. (2024b). To address this issue, we perform a forward simulation for each candidate trajectory $\tau_i$ of the CBV, yielding a rollout $\widetilde{\tau}_i$. The simulation couples a PID controller for trajectory tracking with a kinematic bicycle model for state propagation. The PID controller is identical to that used during closed-loop execution, ensuring behavioral consistency between training and deployment. By evaluating rollouts rather than raw trajectories, we reduce this dynamics gap and obtain more reliable assessments.

In parallel, we also forecast the motions of surrounding actors. During data collection, the current actions $a^{\mathrm{bg}}$ of surrounding actors are recorded. Following the rule-based forecasting scheme in Beißwenger (2024), these actions are assumed constant over the forecast horizon and are used to advance surrounding states. The resulting actor forecasts are combined with the CBV rollouts to compute rewards, thereby ensuring that interaction effects with the environment are faithfully captured in evaluation.

While subsequent rollout is open-loop, the first transition is closed-loop. This step integrates (i) the same PID policy as in real execution, (ii) the observed current actions of surrounding actors, and (iii) a kinematic bicycle model that approximates CARLA's single-step dynamics. Accordingly, the transition from $(s, a, a^{\mathrm{bg}})$ to $s'$ produces a reward consistent with the standard RL structure $(s, a) \rightarrow s' \rightarrow r$. Subsequent rollout steps serve as open-loop estimates of longer-horizon outcomes, enriching evaluation while preserving closed-loop fidelity at the transition boundary.

## B.7 STATE-WISE REWARD MODEL SETUP

To capture diverse human driving styles, we decompose driving behaviors into distinct reward components, following Cusumano-Towner et al. (2025). Different styles are constructed by combining weights assigned to each reward component (detailed in Table 6), enabling a range of behaviors from aggressive to conservative. The total driving reward is defined as:

$$R = R_{\mathrm{collision}} + R_{\mathrm{off\text{-}road}} + R_{\mathrm{comfort}} + R_{\mathrm{lane}} + R_{\mathrm{velocity}} + R_{\mathrm{timestep}}. \tag{17}$$

The individual terms are described as follows:

- $R_{\mathrm{collision}} = -\left(\alpha_{\mathrm{collision}} + |v|\right)\mathbb{1}_{\mathrm{collision}}$: penalizes collisions, with higher penalties at higher speeds.

- $R_{\mathrm{off\text{-}road}} = -\alpha_{\mathrm{boundary}}\mathbb{1}_{\mathrm{boundary}}\,\alpha_{\mathrm{boundary}}$: penalizes deviations from the drivable area.

- $R_{\mathrm{comfort}} = -\alpha_{\mathrm{comfort}}\left(\mathbb{1}_{|a|>4} + \mathbb{1}_{|\omega|>4}\right)$ penalizes excessive acceleration and angular acceleration.

Table 4: Hyperparameters used in *RIFT* Training.

| Parameter | Value |
|---|---|
| Batch size | 256 |
| Rollout buffer capacity | 4096 |
| Fine-tune initial LR | $1 \times e^{-4}$ |
| Minimum LR | $1 \times e^{-6}$ |
| LR decay across iteration | 0.9 |
| LR schedule | Cosine |
| Num. RIFT epoch | 16 |
| Warmup Epoch of RIFT | 3 |
| AdamW weight-decay | $1 \times e^{-5}$ |

Table 5: Hyperparameters of *RIFT* or RL baselines.

| Parameter | Value |
|---|---|
| PPO clipping ratio $\epsilon$ | 0.2 |
| Dual-clip ratio $c$ | 3 |
| Discount factor $\gamma$ | 0.98 |
| $\lambda_{\text{GAE}}$ Schulman et al. (2015) | 0.98 |
| Hidden dimension $D$ | 128 |
| Num. lon. queries $N_{lon}$ | 12 |
| Traj. time horizon $T$ | 80 |
| Map radius | 120m |
| Frame rate | 10Hz |

Table 6: Reward Parameters for Different Driving Styles.

| Parameter | Normal | Aggressive |
|---|---|---|
| $\alpha_{\text{collision}}$ | 20.0 | 5.0 |
| $\alpha_{\text{boundary}}$ | 5.0 | 5.0 |
| $\alpha_{\text{comfort}}$ | 0.8 | 0.8 |
| $\alpha_{\text{l-align}}$ | 0.5 | 0.5 |
| $\alpha_{\text{vel-align}}$ | 0.05 | 0.05 |
| $\alpha_{\text{l-center}}$ | 0.6 | 0.6 |
| $\alpha_{\text{center-bias}}$ | 0.0 | 0.0 |
| $\alpha_{\text{velocity}}$ | 0.1 | 0.2 |
| $\alpha_{\text{timestep}}$ | 0.1 | 0.1 |

- $R_{\text{l-align}} = \alpha_{\text{l-align}} \left( \min\left( \cos\left( \theta_f \right), 0 \right) + \alpha_{\text{vel-align}} \min\left( \cos\left( \theta_f \right) * v, 0 \right) + 0.25 \left( 1 - \frac{|\theta_f|}{\pi/2} \right) \right)$: guides the agent to follow the correct driving direction and remain parallel to the lane markings.

- $R_{\text{l-center}} = -\alpha_{\text{l-center}} \left( \mathbb{1}_{\cos(\theta_f)>0.5} * \left( |x_f - \alpha_{\text{center-bias}}| - \frac{0.05}{\exp(|x_f - \alpha_{\text{center-bias}}| - 0.5)} \right) \right)$: guides the agent to prefer trajectories that remain centered within the lane.

- $R_{\text{velocity}} = \alpha_{\text{velocity}} \max\left( \cos\left( \theta_f \right), 0.0 \right) \mathbb{1}_{3<|v|<20} * |v|$: promotes forward movement and biases the agent toward choosing routes with consistent traffic flow rather than traffic jams.

- $R_{\text{timestep}} = -\alpha_{\text{timestep}} \mathbb{1}_{|v|>0 \vee |a|>0}$ applies a small per-step penalty, encouraging efficiency. It is disabled when the agent is stationary to allow appropriate waiting behavior at intersections.

Building on the reward definitions above, we construct a state-wise reward model $\text{StateWiseRM}(\cdot)$, which computes a scalar reward based on a set of interpretable features extracted from each rollout point $\widetilde{\tau}_i^t$. Specifically, we define a feature extraction function $\phi(\widetilde{\tau}_i^t)$ as:

$$\phi(\widetilde{\tau}_i^t) = \left( \mathbb{1}_{\text{collision}}, \mathbb{1}_{\text{boundary}}, a_{\text{long}}, a_{\text{lat}}, \theta_f, x_f, v, a \right), \tag{18}$$

where:

- $\mathbb{1}_{\text{collision}}$ and $\mathbb{1}_{\text{boundary}}$ are binary indicators of potential collisions and off-road violations;
- $a_{\text{long}}$ and $a_{\text{lat}}$ denote the longitudinal and lateral acceleration;
- $v$ and $a$ are the magnitudes of velocity and acceleration;
- $x_f$ is the lateral distance to the nearest lane centerline;
- $\theta_f$ is the heading deviation concerning the lane direction.

The state-wise reward is then computed as:

$$r_i^t = \text{StateWiseRM}(\phi(\widetilde{\tau}_i^t), s). \tag{19}$$

All features, except the infraction indicators, are directly derived from the rollouts. To estimate future infractions, we follow the forecasting model in Beißwenger (2024) to simulate other agents' future positions based on current states and actions and identify collisions via bounding box overlap. Off-road violations are detected by projecting the rollout trajectory onto the HD map and checking its occupancy relative to the drivable area polygon set.

## B.8 CONTROLLABILITY AND REALISM METRICS

**Kinematic Metrics.** Following Montali et al. (2023), kinematic realism is typically evaluated against ground-truth trajectories. As CARLA provides no expert demonstrations, we adopt distribution-level metrics Chen et al. (2024a); Huang et al. (2025a) to assess CBV behavior in terms of speed and acceleration. Specifically, we employ three measures—the Shapiro–Wilk test on speed (S-SW), the Wasserstein Distance on speed (S-WD), and the Shapiro–Wilk test on acceleration (A-SW)—defined as follows:

- *Wasserstein Distance (WD) Vaserstein (1969):* measures the distance between two distributions $\mu$ and $\nu$. Since CARLA provides a predefined target speed for agents, we use WD to compare the simulated CBV speed distribution with the target speed distribution as the reference.

$$\text{WD}(\mu, \nu) = \inf_{\gamma \in \Pi(\mu, \nu)} \mathbb{E}_{(x,y) \sim \gamma} \left[ d(x, y) \right]. \tag{20}$$

- *Shapiro–Wilk test (SW) Shapiro & Wilk (1965):* evaluates the normality of speed and acceleration distributions—a simplifying assumption supported by empirical traffic studies Zhang et al. (2023a); Yan et al. (2023)—to capture the statistical naturalness of CBV motion.

$$\text{SW} = \frac{\left( \sum_{i=1}^{n} a_i x_{(i)} \right)^2}{\sum_{i=1}^{n} (x_i - \bar{x})^2}, \tag{21}$$

where $a_i$ are coefficients, $x_{(i)}$ are the ordered data points, $x_i$ are the sample values, $\bar{x}$ is the sample mean, and $n$ is the number of data points.

**Interaction Metrics.** Following the metric design principles proposed in the WOSAC challenge Montali et al. (2023) and other widely adopted evaluation frameworks Lin et al. (2024); Chen et al. (2024a), we adopt a set of well-established metrics to comprehensively evaluate agent interactions:

- *Collision Per Kilometer (CPK) Chen et al. (2024a):* the average number of scenario collisions per kilometer of driving distance.

- *Route Progress (RP) Chen et al. (2024a):* the total distance traveled by all CBVs, reflecting route completion.

- *2D Time-to-Collision (2D-TTC) Guo et al. (2023):* the minimum of longitudinal and lateral time-to-collision from the AV's perspective, capturing the interaction risk posed by CBVs.

- *Anticipated Collision Time (ACT) Venthuruthiyil & Chunchu (2022):* a safety-critical metric measuring the AV's proximity to potential collisions, reflecting the interaction intensity introduced by CBVs.

**Map Metrics.** Map metrics evaluate adherence to road geometry, reflecting how well agents remain within drivable areas and comply with map constraints.

- *Off-Road Rate (ORR) Chen et al. (2024a):* the percentage of time CBVs spend off-road on average.

## C  AV EVALUATION DETAILS

### C.1  AV METHODS IMPLEMENTATION

To assess the effectiveness of *RIFT* in generating reliable and interactive scenarios for AV evaluation in the AV-centric closed-loop simulation environment, we evaluate the following representative and stable AV methods:

- *PDM-Lite Beißwenger (2024):* A rule-based privileged expert method that achieves state-of-the-art performance on the CARLA Leaderboard 2.0 by leveraging components such as the Intelligent Driver Model and the kinematic bicycle model. This open-source method serves as a strong baseline for comparison.

- *PlanT Renz et al. (2022):* An explainable, learning-based planning method that operates on an object-level input representation and is trained through imitation learning.

- *UniAD Hu et al. (2023):* A planning-oriented unified framework integrating perception, prediction, mapping, and planning into one end-to-end model using query-based interfaces.

- *VAD Jiang et al. (2023a):* A fast, end-to-end vectorized driving paradigm representing scenes with vectorized motion and map elements for efficient, safe planning.

## C.2 AV EVALUATION METRICS

As detailed in Appendices B.1 and B.4, we develop an AV-centric closed-loop simulation environment, including a training and evaluation pipeline based on Bench2Drive. The AV closed-loop evaluation metrics proposed in Bench2Drive extend the original metrics of the CARLA Leaderboard by emphasizing the specific strengths and weaknesses of different methods across various aspects, such as merging and overtaking, thereby making them suitable for evaluating performance under predefined scenarios. However, as noted in Appendix B.1, replacing predefined scenarios with CBV-generated traffic scenarios precludes the evaluation of specific AV capabilities. To systematically assess the quality of traffic scenarios generated by different CBV methods, we follow the practice of KING Hanselmann et al. (2022) and introduce PDM-Lite Beißwenger (2024)—a rule-based privileged planner—as a reference AV. By measuring its performance under various CBV methods, we evaluate:

- *Feasibility*, via PDM-Lite's Driving Score (DS)—a high DS indicates the PDM-Lite can complete its route without severe collisions or rule violations, implying the generated traffic scenario is feasible.

- *Naturalness*, via Blocked Rate (BR)—a low BR suggests that CBVs do not unrealistically obstruct the AV, reflecting naturalistic behavior.

These metrics enable a principled comparison of traffic quality generated by different CBV methods. Furthermore, to assess the capacity of generated traffic scenarios to expose AV limitations, we test multiple learning-based AV methods under an identical CBV method and quantify their relative performance drop compared to PDM-Lite Beißwenger (2024). The relative driving score degradation ($\Delta DS$) reflects how effectively the traffic scenario stresses the AV policy, with larger drops indicating stronger capability in revealing planning weaknesses.

The evaluation metrics are summarized as follows:

- *Driving Score (DS):* $R_i P_i$ — The main metric of the leaderboard, calculated as the product of route completion and the infraction penalty. Here, $R_i$ represents the percentage of completion of the $i$-th route, and $P_i$ denotes the infraction penalty. The maximum value is 100.

- *Block Rate (BR):* **The number of times the AV remains below** $0.1m/s$ **for over** $3s$**, signaling traffic-induced obstruction. Because BR is affected by both traffic and AV behavior, it is evaluated only under the rule-based PDM-Lite, which avoids self-induced stalls and thus reflects traffic naturalness.**

- *Relative Driving Score Degradation ($\Delta DS$):* The reduction in Driving Score of a learning-based AV compared to PDM-Lite under the same CBV method, indicating how effectively the scenario reveals weaknesses in AV planning.

## C.3 END-TO-END AV VISUALIZATION

To further validate the feasibility of *RIFT* as a closed-loop evaluation framework, we extend its application beyond traffic scenario generation to testing end-to-end autonomous driving algorithms. In contrast to conventional adversarial approaches that often introduce unrealistic or overly aggressive behaviors, *RIFT* generates traffic flows that preserve the realism of human driving styles, engage the autonomous vehicle in genuine interactive behaviors, and maintain feasibility by ensuring that the resulting scenes, though diverse and stress-inducing, remain solvable for the end-to-end method. This balance enables *RIFT* to evaluate robustness under credible conditions while avoiding degenerate or unsolvable scenarios.

Figure 5 presents representative interactions between *RIFT*-generated traffic flows and two representative end-to-end driving models, UniAD and VAD. As shown, *RIFT* adapts seamlessly to different driving policies, producing realistic and interactive scenes where the autonomous vehicle must negotiate with surrounding traffic. These results underscore the capability of *RIFT* to provide realistic yet interactive closed-loop evaluations, highlighting its potential as a versatile tool for testing the robustness of end-to-end AV systems.

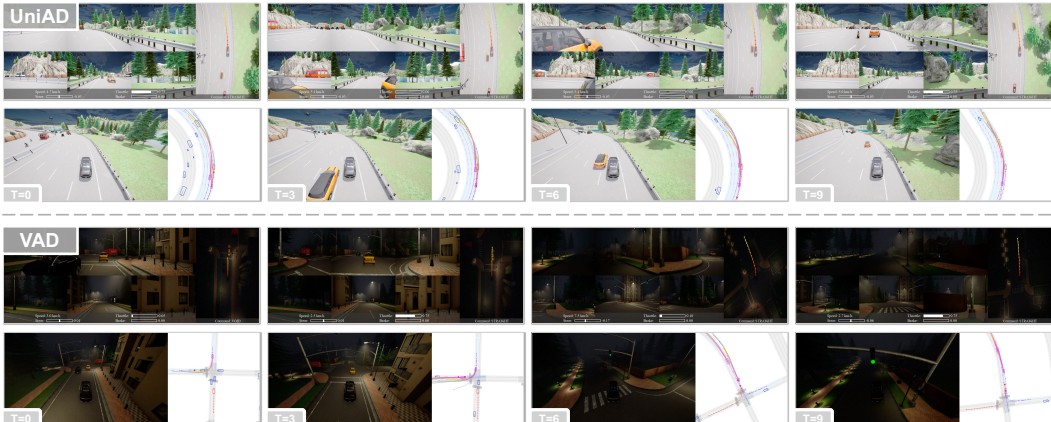

Figure 5: Representative closed-loop interactions between *RIFT*-generated traffic flows and end-to-end autonomous driving algorithms. UniAD (top) and VAD (bottom) are shown interacting with surrounding vehicles orchestrated by *RIFT*, which preserves realistic driving styles while enabling dynamic CBV–AV interactions. The controlled background vehicle (CBV) is highlighted in purple, the autonomous vehicle (AV, end-to-end) in red, and other background vehicles (BVs) in blue.

# D    ADDITIONAL RESULTS

## D.1    DETAILED QUALITATIVE RESULTS OF STYLE-LEVEL CONTROLLABILITY

As discussed in Section 5.4, we investigate the style-level controllability of *RIFT* under different reward configurations. The aggressive variant applies a reduced collision penalty and places greater emphasis on driving efficiency (Table 6), encouraging assertive behaviors such as overtaking. In contrast, the normal configuration imposes a higher collision penalty to promote safer and more conservative driving behaviors.

Quantitative results in Table 3 show that the aggressive variant achieves greater driving efficiency at the expense of more frequent collisions and off-road events. To complement these findings, Figure 6 presents a qualitative comparison in a single-lane intersection scenario where a leading BV halts at a stop sign. The aggressive CBV variant attempts an overtaking maneuver, resulting in a collision, whereas the normal CBV variant yields and waits, demonstrating distinct behavioral patterns induced by different reward preferences. These results highlight the controllability of *RIFT* in modulating driving style according to user-specified reward configuration.

## D.2    DETAILED ANALYSIS IN DRIVING COMFORT

**Metrics.** To further evaluate the driving comfort of different CBV methods, we define several comfort metrics based on Bench2Drive, which assesses agent comfort through acceleration and jerk profiles. Specifically, we measure comfort using the following metrics:

- *Uncomfortable Rate (UCR):* the percentage of simulation time during which CBVs experience discomfort.
- *Driving Jerk (Jerk):* the time derivative of acceleration, quantifying the abruptness of acceleration changes and the smoothness of CBV rollouts.

To determine whether a CBV's current state is considered comfortable, we adopt the Frame Variable Smoothness (FVS) criterion from Bench2Drive:

$$\text{Frame Variable Smoothness (FVS)} = \begin{cases} \text{True} & \text{if lower bound} \leq p_i \leq \text{upper bound.} \\ \text{False} & \text{otherwise} \end{cases} \quad (22)$$

$$p \in \text{smoothness vars}, \ 0 \leq i \leq \text{total frames}$$

The smoothness variables include longitudinal acceleration (expert bounds: [-4.05, 2.40]), maximum absolute lateral acceleration (expert bounds: [-4.89, 4.89]), and maximum jerk magnitude (expert bounds: [-8.37, 8.37]).

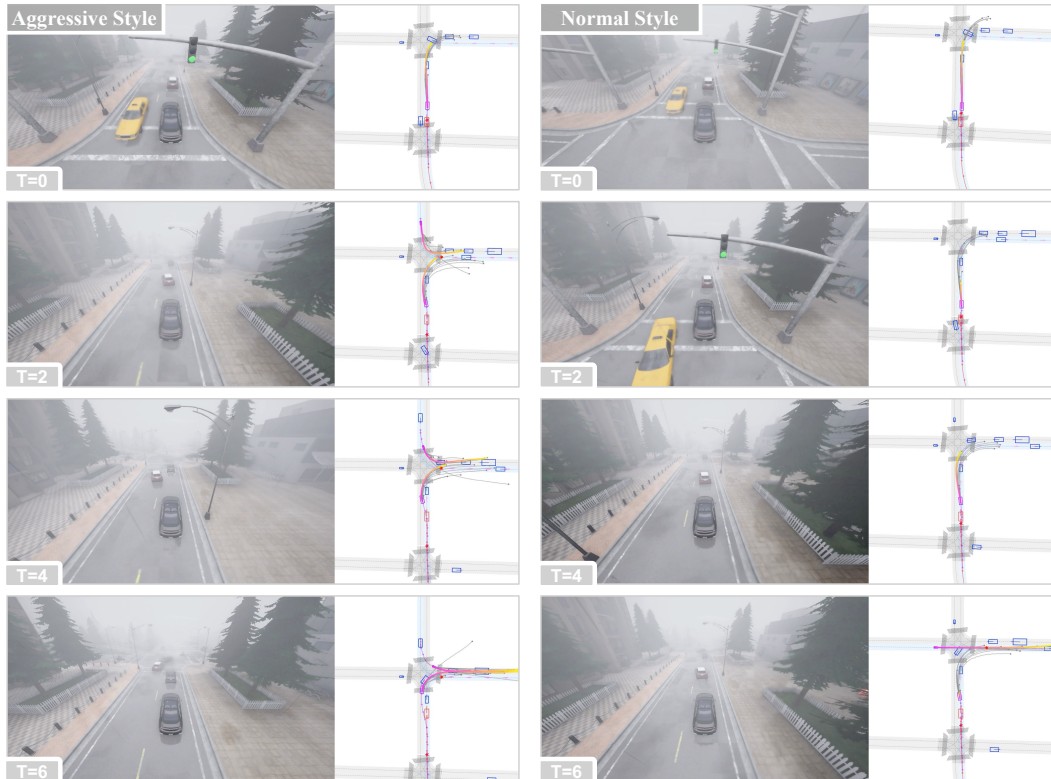

Figure 6: Qualitative illustration of *RIFT*'s style-level controllability under different reward configurations. CBV is marked in purple, AV (PDM-Lite) is in red, and BVs are in blue.

Table 7: **Comparison of CBV Comfort Metrics across Various AV Methods.** Each metric is evaluated across three random seeds.

| Method | PDM-Lite | | PlanT | |
|---|---|---|---|---|
| | UCR ↓ | Jerk ↓ | UCR ↓ | Jerk ↓ |
| Pluto | 56.45 ± 4.14 | -0.16 ± 3.72 | 50.26 ± 2.17 | -0.42 ± 3.38 |
| PPO | 74.76 ± 2.71 | -0.51 ± 4.61 | 74.90 ± 1.21 | 0.40 ± 4.83 |
| FREA | 72.40 ± 1.72 | 0.29 ± 4.61 | 73.48 ± 3.83 | -0.15 ± 4.91 |
| FPPO-RS | 68.33 ± 1.90 | -0.07 ± 3.96 | 66.67 ± 0.82 | -0.15 ± 3.95 |
| SFT-Pluto | 68.14 ± 4.91 | -0.06 ± 4.06 | 59.78 ± 4.72 | -0.11 ± 4.00 |
| RS-Pluto | 70.31 ± 4.07 | 0.32 ± 4.12 | 65.18 ± 2.11 | -0.16 ± 4.07 |
| RTR-Pluto | 55.58 ± 4.76 | -0.19 ± 3.37 | 45.12 ± 2.66 | -0.14 ± 3.34 |
| PPO-Pluto | 58.29 ± 2.70 | -0.32 ± 3.70 | 54.85 ± 5.82 | -0.07 ± 3.40 |
| REINFORCE-Pluto | 68.10 ± 1.22 | 0.23 ± 3.96 | 64.94 ± 5.36 | -0.11 ± 3.96 |
| GRPO-Pluto | 78.58 ± 0.59 | 0.22 ± 4.62 | 77.13 ± 0.65 | -0.23 ± 4.58 |
| RIFT-Pluto (ours) | 76.90 ± 2.82 | 0.59 ± 4.12 | 72.41 ± 4.02 | 0.21 ± 4.44 |

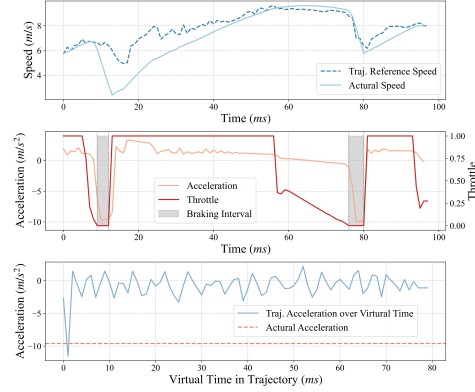

Figure 7: Controller Performance.

**Main Results.** The quantitative results of the comfort metrics are presented in Table 7. All CBV methods exhibit notable levels of driving discomfort. Although the more conservative methods identified in Section 5.2 achieve relatively lower levels of discomfort, a high baseline of discomfort persists between methods.

To investigate the underlying causes of discomfort, we further decouple the planned trajectories from the executed control actions. In CARLA, most CBV methods rely on PID controllers to transform high-level trajectory waypoints into executable driving commands, including throttle, steering, and brake. As shown in Figure 7, the upper panel illustrates the speed tracking curve, while the middle panel presents the raw throttle signal and corresponding acceleration profile.

Because trajectory generation is performed state-wise, predicting only the immediate next action, the reference states may vary discontinuously over time. These discontinuities are amplified by

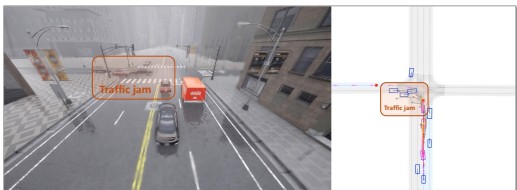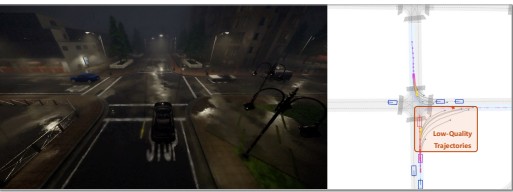

Figure 8: Failure Cases of Route-Conditioned Trajectory Generation.

the PID controller, whose binary throttle/brake responses induce abrupt changes in acceleration, ultimately leading to discomfort during vehicle operation. Such execution-level instabilities are a major contributor to the discomfort observed across CBV methods.

While many CBV methods attempt to mitigate discomfort through fine-tuning strategies that incorporate post-action feedback via reward shaping or expert action alignment, *RIFT* adopts a different approach. It employs a state-wise reward model (see Appendix B.7) that quantifies comfort within the trajectory's virtual forward simulation.

To further analyze this, we visualize both the actual acceleration after executing a selected trajectory and the corresponding virtual-time acceleration (shown in Figure 7). The results reveal that while virtual-time acceleration aligns with actual motion at the beginning of the trajectory, it underestimates acceleration variations in later segments of the trajectory. This leads to an overly conservative estimation of trajectory-level discomfort, resulting in insufficient supervision during training and reflected in *RIFT*'s comfort performance in Table 7.

In summary, the discomfort exhibited by CBV methods can be attributed to two primary sources:

- **Tracking instability**, caused by discontinuities in planned trajectories and the limited control fidelity of PID controllers. Discrete, state-wise planning combined with low-resolution, often binary control outputs amplifies acceleration fluctuations and leads to uncomfortable motion.

- **Inadequate comfort modeling**, particularly in state-wise reward formulations such as that adopted by *RIFT* and GPRO. These formulations fail to capture long-term trajectory-level discomfort, leading to insufficient supervision during training and suboptimal comfort performance.

### D.3 FAILURE CASES ANALYSIS

Figure 8 presents two representative failure modes that reveal structural limitations of our route-conditioned trajectory generator. In the first case (left), the CBV approaches a congested segment where multiple stopped vehicles fully obstruct the forward corridor. Because the candidate set is constructed by expanding along map-provided reference lines, all modalities inevitably lead into the jammed region. With no collision-free alternative available, the CBV resorts to waiting indefinitely, which further amplifies congestion at the scene level.

The second case (right) demonstrates a related but distinct failure pattern. When a blocking vehicle narrows the feasible corridor, most generated candidates drift outside the drivable area or violate kinematic feasibility. As a result, the entire trajectory cluster becomes low-quality, leaving the optimizer with no viable mode to select. In such situations, the CBV can only adopt overly conservative behaviors such as stopping or creeping, not because the optimization fails, but because the underlying candidate space collapses.

Taken together, these cases highlight a core limitation of *RIFT*: its performance is fundamentally bounded by the expressiveness and feasibility of the candidate trajectories. Group-relative optimization is effective only when the trajectory set contains at least one plausible solution; when all modes inherit structural deficiencies from the route-conditioned trajectory generator, no downstream optimization can compensate. Addressing how to elevate trajectory quality and expand feasible coverage without relying on human expert supervision therefore emerges as an important direction for future work.

Figure 9: TTC-Based CBV Identification at Complex Intersections.

### D.4 TTC-BASED CBV IDENTIFICATION

**To explore an alternative to route-overlap–based identification, we evaluate a TTC-based CBV identification variant. For each scene, we compute the time-to-collision (TTC) between the AV and all surrounding agents using constant-velocity extrapolation. Agents whose TTC falls below a predefined threshold are selected as CBVs, after which a global path is generated for each selected vehicle using an A\* global path planning algorithm. This procedure explicitly targets vehicles with potentially high interaction risk and aims to surface non-overlapping cross-traffic agents that may not be captured by route-level heuristics.**

**However, as shown in Figure 9, this approach exhibits notable limitations at complex intersections. The TTC computation assumes straight-line, constant-velocity motion; yet the AV's actual maneuver often deviates significantly from this extrapolation. As a result, some vehicles flagged as "high-risk" do not produce meaningful interactions during closed-loop rollout. These false positives indicate that simply replacing route-overlap rules with safety-metric triggers does not fully resolve the intersection-identification problem.**

**This analysis suggests that robust route-level interaction mining—particularly for multi-lane, multi-phase intersections—is a non-trivial challenge requiring deeper investigation. Future directions include developing intention-aware risk predictors, or VLM-assisted semantic risk analysis that can more reliably identify behaviorally significant agents and infer their corresponding interaction routes in complex urban environments.**

### D.5 VISUALIZATION OF THE AV-CENTRIC CLOSED-LOOP SIMULATION

To qualitatively evaluate the robustness of *RIFT* across diverse AV-centric scenarios, we provide additional temporal visualizations of closed-loop simulations. As shown in Figure 10, the traffic scene consists of the autonomous vehicle (AV, controlled by PDM-Lite), background vehicles (BVs), and critical background vehicles (CBVs), which interact dynamically over time.

The visualizations demonstrate the ability of *RIFT* to generate temporally coherent, realistic, and controllable trajectories across a variety of traffic situations. Even under complex and evolving closed-loop conditions, *RIFT* maintains stable multimodal behavior, highlighting its effectiveness in simulating realistic and controllable traffic scenarios around the AV.

## E DISCUSSION AND BROADER IMPLICATIONS

### E.1 LIMITATIONS AND FUTURE WORK.

In the current framework, the generation head is frozen during RL fine-tuning, and its reliability stems from the robust trajectory generation capability learned during IL pre-training. However, without reliable expert demonstrations, the realism and robustness of generated trajectories cannot be further improved during RL fine-tuning. This limitation highlights a key avenue for future research: developing methods—potentially leveraging RL or other self-improvement paradigms—that can enhance the trajectory generation quality without relying on expert demonstrations.

### E.2 USE OF LARGE LANGUAGE MODELS (LLMS)

The large language model (LLM) was employed as a general-purpose writing assistant during the preparation of this manuscript. Its use was limited to:

- Language refinement: improving grammar, syntax, and overall readability to ensure clarity and professionalism.

- Style adjustments: suggesting more concise and precise phrasing while preserving the original meaning and technical content.

The LLM was not involved in research ideation, experimental design, data collection, analysis, or interpretation of results. All intellectual contributions and scientific conclusions are solely those of the authors. This disclosure is provided in accordance with the conference guidelines on LLM usage.

### E.3 SOCIAL IMPACT

**Positive Societal Impacts.** This work presents a practical framework that bridges the gap between realism and controllability in traffic simulation. By decoupling pre-training and fine-tuning, our method enables models pre-trained on real-world datasets to adapt effectively to physics-based simulators, preserving trajectory-level realism and route-level controllability while improving long-horizon closed-loop performance. This paradigm establishes a viable pathway for transitioning data-driven approaches to physics-based simulators, enabling more reliable closed-loop testing and training. Consequently, it advances safer and more robust autonomous systems.

**Negative Societal Impacts.** While fine-tuning in physics-based simulators improves closed-loop performance, it may also lead to overfitting to the specific characteristics of the simulator. As a result, the learned policy could struggle to generalize beyond the simulated environment, giving rise to a sim-to-real gap. This gap poses challenges for real-world deployment, as models that perform well in simulation may not retain the same level of reliability when applied to actual autonomous driving systems. Such discrepancies can affect the testing and training stages, highlighting the need for further work to ensure real-world transferability.

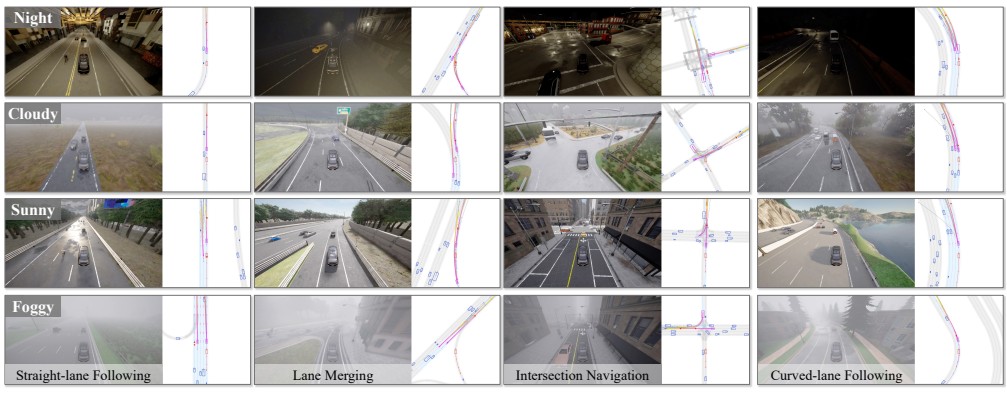

(a) Robustness of *RIFT* across diverse AV-centric traffic scenarios.

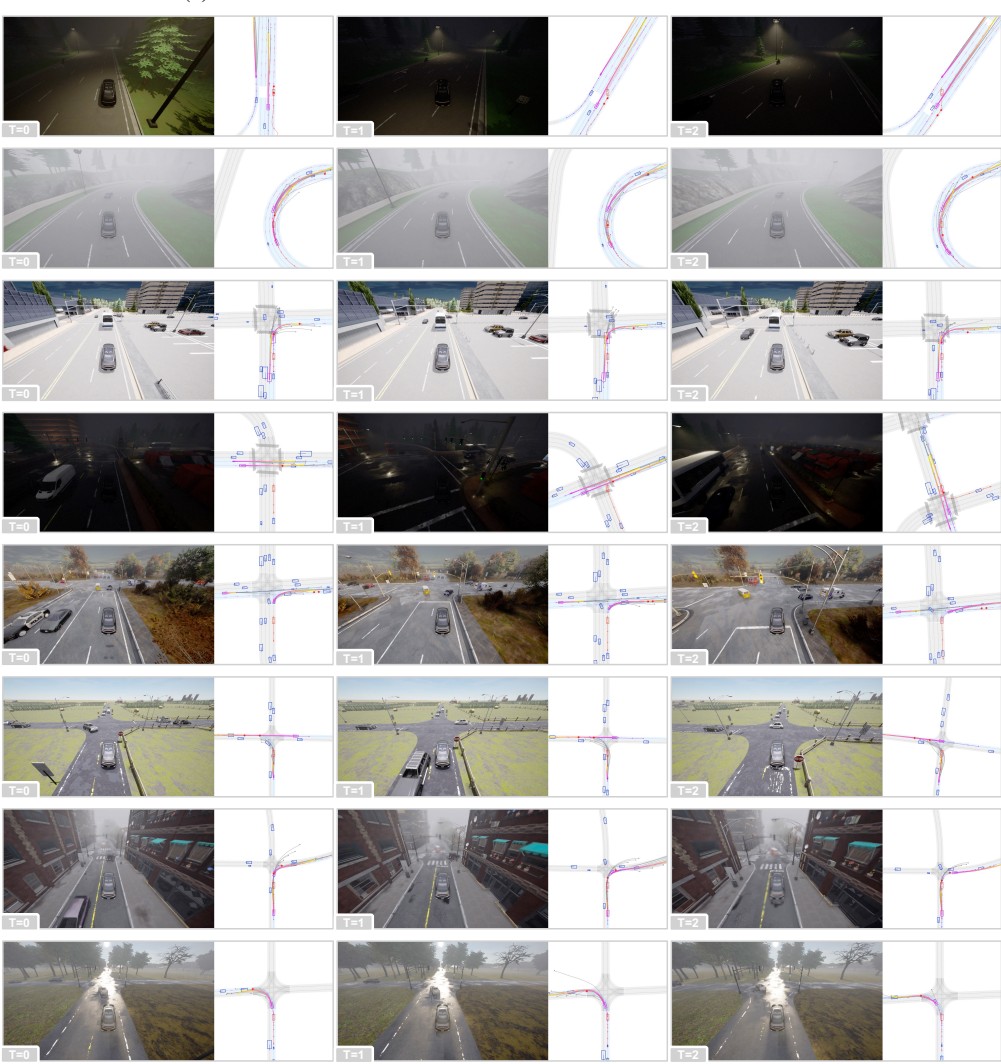

(b) Temporal stability of *RIFT* in closed-loop simulation.

Figure 10: Visualizations of *RIFT* in diverse AV-centric scenarios. (a) Robustness of *RIFT* across diverse AV-centric traffic scenarios. (b) Temporal stability of *RIFT* in closed-loop simulation. CBV is marked in purple, AV (PDM-Lite) is in red, and BVs are in blue.

