# OpenReview forum: "RIFT: Group-Relative RL Fine-Tuning for Realistic and Controllable Traffic Simulation"
_ICLR.cc/2026/Conference — Submitted to ICLR 2026_

### Official Review · Reviewer_Bo1Q · 2025-10-16

**Soundness:** 3
**Presentation:** 3
**Contribution:** 3
**Rating:** 6
**Confidence:** 4

**Summary:**

The paper introduces a new method to indentify the key background vehicle and improving the vehicle's trajectory realism by group-relative RL finetuning. The method preverses the trajectory-level realism and multi-modality. Experiments demonstrates RIFT can improve realism and controllability in traffic simulation.

**Strengths:**

1.The appendix is detailed and the supplementary videos is comprehensive and vivid.
2. The experiment includes extensive prior works with reasonable realism metrics.
3. The paper is well written and easy to follow.

**Weaknesses:**

1. The method to indentify the CBV is problematic because it cannot indentify the vehicle in the intersection which has different routes with the ego vehicle but has large collision risk. It is demonstrated in the videos.
2. The method to evaluate the realism in the simple CARLA environment where all BV behaves convervatively. Consider evaluate and compare in the Waymo Sim Agent benchmark where groud-truth is provided for better evaulation of realism.

**Questions:**

1. Why does the other method except PDM-lite in table 2 not report the BR metrics?
2. How do you refine the all head? What is the loss for finetuning trajectory generation head?

---

> ### Author Response · Authors · 2025-11-20
>
> We thank the reviewer for the detailed comments and constructive questions. To provide a clear and coherent response, we have grouped the comments by theme and address each set of questions in turn. This organization allows us to clarify the intended scope of our method, explain our design choices, and provide the necessary experimental evidence and technical details. The following sections present our point-by-point responses.
>
> **Q1:  CBV Identification**
>
> > The method to indentify the CBV is problematic because it cannot indentify the vehicle in the intersection which has different routes with the ego vehicle but has large collision risk. It is demonstrated in the videos.
>
> We thank the reviewer for pointing out this case. The intersection example in the video is indeed a situation where our current CBV identification may fail to select a vehicle that has high collision risk but does **not** share a route with the AV. This behavior is consistent with the design scope of our method: **RIFT is not a safety-critical scenario generation framework whose goal is to capture all potentially dangerous events.** Instead, our objective is to introduce **realistic, controllable interactions around the AV at the route level**, so that the AV encounters plausible interacting traffic while we preserve human-like behavior and enable style control.
>
> With this goal in mind, our current CBV identification is intentionally defined at the **route-interaction level**: we focus on vehicles whose planned routes overlap with the AV’s route, and then treat them as candidates for route-conditioned CBV planning. This rule-based scheme is **simple, fast, and fully automatic**, and it guarantees that the selected CBVs have clear, route-level interaction semantics with the AV. We acknowledge that this design is **conservative from the viewpoint of safety-critical coverage**: it can miss certain cross-traffic vehicles at intersections that do not share a route with the AV but still pose a high instantaneous risk (as in the reviewer’s example).
>
> Importantly, however, **CBV identification is decoupled from route-conditioned CBV planning** in our framework. The contribution of RIFT lies primarily in the latter—i.e., learning realistic and controllable CBV behaviors once a set of CBVs has been selected. The current identification module should therefore be viewed as a **stable, reliable baseline mechanism** for introducing route-level interactions, rather than as an exhaustive detector of all safety-critical situations. Thanks to this modularity, the identification step can be upgraded without changing our planning pipeline.
>
> In line with this, in addition to the main distance-based identification used throughout the paper, we evaluate a safety-critical identification variant as an ablation. We added a variant that applies a **TTC < 2 s filter** when selecting CBVs, so that only vehicles with sufficiently small time-to-collision are controlled. As shown below, this TTC-based variant yields slightly lower 2D-TTC and ACT values during evaluation, indicating that the generated traffic indeed becomes more safety-critical and interaction-intensive:
>
> |                                      |   2D-TTC    |     ACT     |
> | :----------------------------------: | :---------: | :---------: |
> | CBV Identification w/ TTC Filtering  | 2.69 ± 1.33 | 2.63 ± 1.36 |
> | CBV Identification w/o TTC Filtering | 2.74 ± 1.30 | 2.71 ± 1.32 |
>
> These results suggest that **risk-aware CBV identification can be naturally integrated into our framework**, and we see this as a promising direction for future work. For example, TTC/ACT-based rules, more advanced safety-critical mining techniques, or even **VLM-assisted semantic risk analysis** could be plugged into the identification stage to capture a broader spectrum of dangerous interactions, while our route-conditioned CBV planning module continues to provide realistic and controllable behaviors for the selected CBVs.

---

> > ### Comment · Reviewer_Bo1Q · 2025-11-22
> > **Reponse**
> >
> > I think the author misunderstands my meaning. I not mean that the RIFT should generate safety-critical scenario generation. I just take the vehicle at interaction as an example, which have dense inteactions with ego vehicle but are not controlled. This is unreasonable. Why do you only focus on vehicles whose planned routes overlap with the AV’s route instead of focusing on vehicles with possible interactions?

---

> > > ### Comment · Reviewer_Bo1Q · 2025-11-22
> > > **Reponse**
> > >
> > > In your ABLATION STUDY (Q3), you said you refine the all head. How do you achive this?

---

> > > > ### Author Response · Authors · 2025-11-22
> > > >
> > > > In this ablation, the **loss is exactly the same** for “Scoring Head” and “All Head”: we only use the main RL objective in Eq. (9), and **do not add any supervised loss on the generation head**, since in closed-loop we do not have reliable ground-truth trajectories. The only difference is which parts are unfrozen: in “Scoring Head” we freeze the generation head and only update the scoring head; in “All Head” we unfreeze both and let Eq. (9) backpropagate through the whole head. The results indicate that, **without reliable closed-loop trajectory supervision, freezing the generation head and only fine-tuning the scoring head better preserves the generative capability learned during pretraining**.

---

> > > > > ### Comment · Reviewer_Bo1Q · 2025-11-23
> > > > > **Reponse**
> > > > >
> > > > > 1. I think it is important to emphasize these limitations now and analyze the influence of the hard cap on the number of CBVs per timestep on the simulation realism.  If you have time, it will be better to add the comparision with the more comprehensive interaction-mining strategies. I think the main problem is that the route ignores the intersection node.
> > > > > 2. Let Eq. (9) backpropagate through the whole head. I think the pi(theta) is only related to the score head. How do you make it update the generation head. Please making it more clear.
> > > > > 3. In Q2, "trajectory realism is inherited from the IL pretraining" is wrong. The IL only learns the trajectory distribution only in the expert state distribution not in the policy state distribution.
> > > > > 4. Waymo realism has more realism metrics than speed and acceleration distributions. It is better to compare the metametric.
> > > > > 5. It is better to add the explain of 'Why does the other method except PDM-lite in table 2 not report the BR metrics?" to the paper and report the BR metrics in the appendix.

---

> > > > > > ### Author Response · Authors · 2025-11-25
> > > > > >
> > > > > > **Q1: CBV Identification & Interaction-Mining**
> > > > > >
> > > > > > > I think it is important to emphasize these limitations now and analyze the influence of the hard cap on the number of CBVs per timestep on the simulation realism. If you have time, it will be better to add the comparision with the more comprehensive interaction-mining strategies. I think the main problem is that the route ignores the intersection node.
> > > > > >
> > > > > > We thank the reviewer for the insightful follow-up comments. As suggested, we conducted additional analyses to clarify the limitations of our CBV identification module.
> > > > > >
> > > > > > **(1) Impact of the hard cap on CBV count.**
> > > > > > We performed an ablation varying the maximum number of CBVs per timestep. Increasing this cap does not noticeably affect motion-level realism, indicating that the individual CBV behavior model remains stable. However, collision rate (CPK) and off-road rate (ORR) increase slightly when too many CBVs are controlled concurrently, indicating that excessive AV–CBV interactions increase the likelihood of scene-level violations.
> > > > > >
> > > > > > |               | JSD Speed (↓) | JSD Acc (↓) | JSD TTC (↓) | S-SW (↑) | S-WD (↓) | A-SW (↑) | CPK (↓) | ORR (↓) |
> > > > > > | :-----------: | :-----------: | ----------- | ----------- | :------: | :------: | :------: | :-----: | :-----: |
> > > > > > | CBV Limit = 2 |     0.08      | 0.26        | 0.03        |   0.97   |   4.46   |   0.93   |  6.83   |  0.36   |
> > > > > > | CBV Limit = 4 |     0.08      | 0.25        | 0.03        |   0.96   |   4.59   |   0.94   |  8.25   |  0.77   |
> > > > > >
> > > > > > **(2) Comparison with more comprehensive interaction-mining strategies.**
> > > > > >
> > > > > > In response to the reviewer’s suggestion, we experimented with a more comprehensive interaction-mining strategy that focuses on vehicles with potential interactions. Specifically, we adopted a commonly used safety-critical indicator (TTC) to identify high-risk vehicles and then constructed their interaction routes under the standard constant-velocity assumption. This design directly follows the reviewer’s intention of prioritizing vehicles that may strongly interact with the AV. Detailed results are reported in Appendix D.4.
> > > > > >
> > > > > > However, our experiments show that—although this TTC-based approach can partially capture non-overlapping intersection cases—it also introduces new issues. As illustrated in Fig. 9, at complex junctions the AV’s actual maneuver often deviates from the constant-velocity extrapolation underlying TTC, leading to cases where vehicles flagged as “high-risk” do not produce meaningful interactions in practice. This suggests that simply replacing route-overlap heuristics with safety-metric triggers does not fully resolve the intersection-identification problem.
> > > > > >
> > > > > > We therefore believe that **robust route-level interaction mining for complex intersections is a non-trivial research problem** that requires more substantial investigation beyond the scope of this work. Our exploration confirms the reviewer’s insight, while also showing that addressing this challenge reliably will require future, dedicated research.
> > > > > >
> > > > > > **(3) Why these limitations do not affect our core contribution.**
> > > > > > We emphasize that CBV identification is decoupled from route-conditioned CBV planning. RIFT’s main contribution lies in learning realistic and controllable CBV behaviors once a set of CBVs and interaction routes are given. Our current identification is intended as a simple, stable baseline to automate route-level interactions, and can be replaced by more advanced mining strategies without modifying the rest of the pipeline.
> > > > > >
> > > > > > We have clarified these limitations and future extensions in Appendix B.2.
> > > > > >
> > > > > > **Q2: Backprop Through Whole Head**
> > > > > >
> > > > > > > Let Eq. (9) backpropagate through the whole head. I think the pi(theta) is only related to the score head. How do you make it update the generation head. Please making it more clear.
> > > > > >
> > > > > > In both “Scoring Head’’ and “All Head’’ settings, the main learning objective is the RL loss in Eq. (9). We emphasize that we do not use any trajectory-level supervised targets for the generation head, since in closed-loop we do not have reliable ground-truth trajectories. The implementation includes a light yaw-norm regularizer inherited from Pluto, but this term is a pure geometric regularization, not a source of trajectory supervision, and does not alter the fact that the generator has no reliable closed-loop supervision.
> > > > > >
> > > > > > When the generation head is unfrozen, this weak regularization introduces minor gradients, but the signal is too limited to provide meaningful guidance in closed-loop rollouts and does not improve generation quality. Empirically, this variant does not outperform the simpler configuration. Freezing the generation head therefore better preserves the IL-pretrained generative prior, while Eq. (9) cleanly fine-tunes only the scoring head.

---

> > > > > > ### Author Response · Authors · 2025-11-25
> > > > > >
> > > > > > **Q3: IL Trajectory Realism & Covariate Shift**
> > > > > >
> > > > > > > In Q2, "trajectory realism is inherited from the IL pretraining" is wrong. The IL only learns the trajectory distribution only in the expert state distribution not in the policy state distribution.
> > > > > >
> > > > > > We appreciate the reviewer’s feedback on the covariate shift issue. As pointed out, IL typically learns from the expert state distribution, and transitioning to the policy state distribution can lead to covariate shift, where the observed states during training differ from those encountered during execution. This is a well-known issue in IL, and we fully acknowledge it.
> > > > > >
> > > > > > The core aim of our work is to mitigate the effects of covariate shift in a new closed-loop simulation environment, where reliable expert trajectories are unavailable. In this context, while covariate shift is inherent in IL, our approach minimizes its impact by leveraging reliable prior information—specifically, **vectorized map and reference lines** during IL pretraining. These conditions are reliable and consistent in the simulation environment, allowing the trajectory generation head to remain more robust to covariate shift compared to the trajectory scoring head, which is more sensitive to such shifts.
> > > > > >
> > > > > > Thus, by decoupling the trajectory generation and scoring processes, we ensure that the IL pretraining phase preserves the route-conditioned generation capabilities, which are less susceptible to covariate shift. RL fine-tuning is then focused on the scoring head, which is more influenced by the distribution shift, effectively alleviating the impact of covariate shift in the absence of reliable trajectory-level supervision.
> > > > > >
> > > > > > Further analysis of covariate shift and how our design addresses it is provided in Section 5.2 of the paper.

---

> > > > > > ### Author Response · Authors · 2025-11-25
> > > > > >
> > > > > > **Q4: Realism Metrics & WOSAC Comparison**
> > > > > >
> > > > > > > Waymo realism has more realism metrics than speed and acceleration distributions. It is better to compare the metametric.
> > > > > >
> > > > > > We appreciate the reviewer’s suggestion to evaluate realism using the broader metric suite from the Waymo Sim Agents Challenge (WOSAC). We conducted a detailed analysis of all WOSAC realism metrics and found that **most of them rely on real-world HD maps and  distributional statistics of Waymo scenarios**, which are not available or directly comparable in CARLA/Bench2Drive. Using them would introduce **scene-dependent bias rather than policy-dependent realism**. Specifically:
> > > > > >
> > > > > > - **Angular speed / angular acceleration** primarily reflect underlying road curvature statistics in Waymo’s real-world logs. CARLA maps differ significantly in curvature frequency and topology, causing these metrics to conflate map geometry differences with behavioral differences.
> > > > > > - **Distance to nearest object** and **distance to road edge** require HD maps and real-world object layouts; their values shift dramatically when map resolution or lane-boundary definitions differ, making cross-platform comparison invalid.
> > > > > > - **Collision Indication and Off-Road Indication** are already captured by our CPK and ORR metrics in Table 1.
> > > > > >
> > > > > > After excluding metrics dominated by map or scenario differences, **linear speed**, **linear acceleration**, and **TTC** remain the most **platform-agnostic** and reliably comparable indicators of motion realism. These metrics are also used in prior simulation work without real-log replay (e.g., D2RL [1], RTR [2]).
> > > > > >
> > > > > > Following these standards, we compute the JSD between generated and real distributions for speed, acceleration, and TTC. Under these cross-platform-valid realism metrics, RIFT consistently matches or surpasses all baselines, demonstrating more realistic motion behavior within the constraints of CARLA simulation environment.
> > > > > >
> > > > > > |      Method       |  JSD Speed (↓)  |   JSD Acc (↓)   |   JSD TTC (↓)   |
> > > > > > | :---------------: | :-------------: | :-------------: | :-------------: |
> > > > > > |       pluto       |   0.10 ± 0.00   |   0.24 ± 0.01   |   0.05 ± 0.04   |
> > > > > > |        ppo        |   0.09 ± 0.01   |   0.30 ± 0.02   |   0.08 ± 0.00   |
> > > > > > |       frea        |   0.08 ± 0.02   |   0.27 ± 0.01   |   0.05 ± 0.00   |
> > > > > > |      fppo_rs      |   0.12 ± 0.01   |   0.30 ± 0.02   |   0.05 ± 0.01   |
> > > > > > |     sft-pluto     |   0.13 ± 0.01   |   0.26 ± 0.01   |   0.09 ± 0.02   |
> > > > > > |     rs-pluto      |   0.10 ± 0.01   | **0.24 ± 0.00** |   0.07 ± 0.00   |
> > > > > > |     rtr-pluto     |   0.15 ± 0.02   |   0.28 ± 0.01   |   0.04 ± 0.02   |
> > > > > > |     ppo-pluto     |   0.09 ± 0.01   | **0.24 ± 0.00** |   0.04 ± 0.01   |
> > > > > > |  reinforce-pluto  |   0.11 ± 0.01   |   0.24 ± 0.01   |   0.06 ± 0.01   |
> > > > > > |    grpo-pluto     |   0.10 ± 0.01   |   0.27 ± 0.00   |   0.04 ± 0.00   |
> > > > > > | RIFT-pluto (ours) | **0.08 ± 0.00** |   0.26 ± 0.01   | **0.03 ± 0.00** |
> > > > > >
> > > > > > [1] Dense reinforcement learning for safety validation of autonomous vehicles. Nature 2024.
> > > > > >
> > > > > > [2] Learning Realistic Traffic Agents in Closed-loop. CoRL 2023.
> > > > > >
> > > > > > **Q5: BR Metric Reporting**
> > > > > >
> > > > > > > It is better to add the explain of 'Why does the other method except PDM-lite in table 2 not report the BR metrics?" to the paper and report the BR metrics in the appendix.
> > > > > >
> > > > > > Thanks for the suggestion. We have added a clarification on the BR metric in Section 5.3 and Appendix C.2.

---

> > > ### Author Response · Authors · 2025-11-22
> > >
> > > Thank you for the clarification, and we apologize for previously misunderstanding your point. We fully agree that, in dense intersection scenes, our current CBV identification may miss agents that do interact with the ego vehicle. This mainly comes from two design choices.
> > >
> > > First, we impose a hard cap on the number of CBVs per timestep (typically 2) to avoid controlling too many interacting vehicles and thereby creating overly congested or unrealistic scenes. Once this budget is filled, other potentially interacting vehicles cannot be promoted to CBVs; this behavior can be mitigated by increasing this hyperparameter. Second, we currently adopt a simple route-level interaction rule that selects vehicles whose planned routes overlap with the ego route. As you pointed out, this conservative criterion can indeed overlook cross-traffic vehicles whose routes do not overlap but can still come very close and generate strong interactions with the ego. We will explicitly state these limitations of our current CBV identification module in the revised version.
> > >
> > > A natural remedy is to incorporate more comprehensive interaction-mining strategies, similar in spirit to MixSim[1]: decoupling interaction and realism by (i) searching for interactive routes that maximize conflict with the ego (e.g., via minimal distance-to-collision) to identify agents that are likely to have strong interactions, and then (ii) using our route-conditioned policy to control the behavior of these selected CBVs in a realistic manner.
> > >
> > > We will add this discussion and highlight such interaction-based identification as an important and compatible extension direction of RIFT. We again thank you for this insightful comment, which helped us better articulate both the limitations and future evolution of our CBV identification design.
> > >
> > > [1] Suo S, Wong K, Xu J, et al. Mixsim: A hierarchical framework for mixed reality traffic simulation. CVPR 2023.

---

> ### Author Response · Authors · 2025-11-20
>
> **Q2: Realism Evaluation:**
>
> > The method to evaluate the realism in the simple CARLA environment where all BV behaves convervatively. Consider evaluate and compare in the Waymo Sim Agent benchmark where groud-truth is provided for better evaulation of realism.
>
> We appreciate the reviewer’s concern regarding realism. In fact, **trajectory realism is inherited from the IL pretraining**, which remains frozen during RL finetuning in RIFT. Our method is explicitly designed to preserve the trajectory-generation realism learned from large-scale datasets, while allowing RL to refine high-level behavior and mitigate closed-loop distribution shift. Importantly, this is consistent with the standard practice in WOSAC, where IL is the primary mechanism for learning realistic and multi-modal trajectory distributions.
>
> RIFT is conceptually compatible with IL-pretrained models that provide trajectory-generation and scoring modules, including those trained for WOSAC. We adopt Pluto in our experiments because its IL-pretrained trajectory generation module has been demonstrated in prior work[1] to produce realistic, multi-modal trajectories, making it a suitable baseline for studying our contributions. Our focus is therefore on improving controllability and closed-loop robustness through RL fine-tuning without compromising trajectory-level realism.
>
> At the same time, we fully agree with the reviewer that empirical comparison to real-world driving statistics is valuable. Following InfGen[2], we extract the global speed and acceleration distributions from the Waymo Open Motion Dataset and **compute the JSD between the generated distributions and the real distributions**. The results show that RIFT matches the real speed distribution more closely than competing methods, and achieves comparable performance on acceleration distribution.
>
> |      Method       |  JSD Speed (↓)  |   JSD Acc (↓)   |
> | :---------------: | :-------------: | :-------------: |
> |       pluto       |   0.10 ± 0.00   |   0.24 ± 0.01   |
> |        ppo        |   0.09 ± 0.01   |   0.30 ± 0.02   |
> |       frea        |   0.08 ± 0.02   |   0.27 ± 0.01   |
> |      fppo_rs      |   0.12 ± 0.01   |   0.30 ± 0.02   |
> |     sft-pluto     |   0.13 ± 0.01   |   0.26 ± 0.01   |
> |     rs-pluto      |   0.10 ± 0.01   | **0.24 ± 0.00** |
> |     rtr-pluto     |   0.15 ± 0.02   |   0.28 ± 0.01   |
> |     ppo-pluto     |   0.09 ± 0.01   | **0.24 ± 0.00** |
> |  reinforce-pluto  |   0.11 ± 0.01   |   0.24 ± 0.01   |
> |    grpo-pluto     |   0.10 ± 0.01   |   0.27 ± 0.00   |
> | RIFT-pluto (ours) | **0.08 ± 0.00** |   0.26 ± 0.01   |
>
> **Q3: Experiment Clarification**
>
> > Why does the other method except PDM-lite in table 2 not report the BR metrics?
>
> In Bench2Drive, BR is computed by checking whether the AV’s speed drops below **0.1 m/s for more than 3 seconds**, which is treated as an indicator that the AV is “blocked” by surrounding traffic. BR is influenced by **both the planner’s capability and the realism of the surrounding traffic flow**.
>
> We report BR only for the rule-based planner because, under our all-green-light setting, it moves continuously except in brief emergency moments. Thus, any blockage for the rule-based baseline can be attributed solely to the traffic flow, making BR a valid indicator of traffic realism.
>
> For learning-based planners, however, BR cannot be interpreted this way: they often incur self-induced blocking due to imperfect policies, making it impossible to separate planner errors from traffic congestion. Therefore, BR from learning-based methods is not meaningful for evaluating scenario realism, which is why we do not report it.
>
> > How do you refine the all head? What is the loss for finetuning trajectory generation head?
>
> Our core idea is to **preserve the strong trajectory-generation capability learned during IL pretraining while enabling RL to refine only the high-level decision behavior**. To achieve this, we adopt a structurally disentangled design: **the trajectory-generation head is frozen, and only the scoring head is finetuned**. This allows RL to adjust trajectory selection and behavioral style without overwriting the low-level motion fidelity obtained from IL.
>
> Accordingly, the trajectory-generation head is trained exclusively during IL pre-training using the standard regression loss for future-trajectory planning, consistent with conventional IL practice. It remains fixed during RL finetuning.
>
> [1] Cheng J, Chen Y, Chen Q. Pluto: Pushing the limit of imitation learning-based planning for autonomous driving. arXiv 2024.
>
> [2] Yang X, Tan S, Krähenbühl P. Long-term traffic simulation with interleaved autoregressive motion and scenario generation. ICCV 2025.

---

### Official Review · Reviewer_WoWw · 2025-10-29

**Soundness:** 2
**Presentation:** 2
**Contribution:** 2
**Rating:** 2
**Confidence:** 5

**Summary:**

This paper proposes **RIFT**, a closed-loop reinforcement learning fine-tuning framework for driving models. The method pre-trains on NuPlan and fine-tunes using **GRPO** (a PPO variant without KL-regularization, with dual clipping) in the CARLA simulator. The goal is to improve controllability and realism in closed-loop driving.

Although the authors provide many experiments, I found the terminology and takeaways unclear (see weaknesses below), which makes it hard to judge the effectiveness of RL-finetuning in this work.

**Strengths:**

- The qualitative results are comprehensive and good for understanding the performance
- I appreciate authors provide several baselines, but the key baselines, such as CAT-K or Waymo Sim Agent Challenge would be more insightful.

**Weaknesses:**

- This works proposes a different metrics compared to Waymo Sim Agent Challenge, I suggest the authors either take 1-2 baselines from Sim Agent Challenge and evaluate their settings or, simply adapt RIFT for Waymo Sim Agent Challenge. Currently is hard for reviewers to understand what are the strength and limitations of RIFT just by looking at  table numbers.
- For AV Evaluation, it is hard to draw insights from Table 2 since there are two factors (Sim Agents and different planners). For example, RIFT might be too reactive that gives wrong esimate or the best performance for AV planners.

**Questions:**

- What is the main advantage of using CARLA/Meta Drive as a fine-tuning simulator? What new aspects does it provide? More accurate dynamics? These simulators also provide an approximation of dynamics, and traffic simulation usually focuses on high-level behavior instead of low-level control.

- There are many works that focus on GRPO fine-tuning of traffic models, e.g., [1][2]. However, the main advantage of the RIFT in this work remains unclear. For example, the authors can consider: 1) What new aspect does it unlock for evaluating planning algorithms?

[1] Wang M., Wang J., Ye T., Chen J., Yu K. (2025). *Do LLM Modules Generalize? A Study on Motion Generation for Autonomous Driving.* CoRL.
[2] Ahmadi E., Schofield H. (2025). *RLFTSim: Multi-Agent Traffic Simulation via Reinforcement Learning Fine-Tuning.* Technical Report for the Waymo Open Sim Agents Challenge.

---

> ### Author Response · Authors · 2025-11-20
>
> We thank the reviewers for their careful reading and valuable feedback. Below we provide point-by-point responses to all questions.
>
> **Q1: Novelty of RIFT**
>
> > There are many works that focus on GRPO fine-tuning of traffic models, e.g., [1] [2]. However, the main advantage of the RIFT in this work remains unclear. For example, the authors can consider: 1) What new aspect does it unlock for evaluating planning algorithms?
>
> We appreciate the reviewer’s thoughtful questions. Our key clarification is that **RIFT addresses a different problem from prior GRPO-based traffic-model finetuning works (e.g., RLFTSim, LLM2AD)**, which explains why our goals, metrics, and simulator choice necessarily diverge from the Waymo Sim Agents Challenge (WOSAC).
>
> (a) What is new compared to prior GRPO-based traffic models?
>
> Prior works use GRPO to calibrate traffic models toward the *naturalistic distribution* of logged human driving. Their main objective is **unconditional realism**—matching offline feature histograms on vectorized trajectories—and they have already shown that group-relative RLFT is effective for this class of tasks.
>
> **RIFT unlocks a different capability:** we focus on **AV-centric, interaction-aware, realistic and controllable traffic simulation for evaluating modern closed-loop AV planners**. Our objective is not to mimic unconditional natural traffic in isolation, but to create **realistic, interaction-rich, and controllable traffic around a specific AV route and planner**, and to study how different AV planners behave under such interactive conditions. This AV-centric, closed-loop perspective is **not the design focus of WOSAC**, which primarily assesses unconditional multi-agent realism.
>
> Conceptually, many WOSAC-style models can be viewed as providing **strong IL-pretrained baselines** (good unconditional realism in a vectorized log domain). RIFT is designed as a **second-stage, closed-loop fine-tuning framework** that starts from such IL-pretrained baseline (here, Pluto on NuPlan) and improves **interaction controllability and AV-centric closed-loop evaluation** in a high-fidelity simulator. In that sense, our contribution is complementary to Sim Agents: we build on IL realism rather than re-solving it from scratch.
>
> (b) Relation to standard GRPO formulation
>
> Finally, we note that **RIFT does not directly apply the standard GRPO formulation used in prior works.** Our setting differs significantly: the trajectory-generation head is frozen, so the set of candidate CBV trajectories is **fixed, small, and highly informative**. Since no new modes are sampled or generated, weighting candidates by the old-policy distribution—as in standard GRPO—would collapse multimodality and suppress useful but low-probability modes. Instead, we use a **uniformly weighted group-relative advantage** over the exhaustively enumerated candidates, allowing the policy to directly optimize the within-group ranking while **preserving multimodality and avoiding within-group sampling bias**.

---

> ### Author Response · Authors · 2025-11-20
>
> **Q2: Relation to WOSAC & Why RIFT Is Evaluated in CARLA Rather Than WOSAC**
>
> > This works proposes a different metrics compared to Waymo Sim Agent Challenge, I suggest the authors either take 1-2 baselines from Sim Agent Challenge and evaluate their settings or, simply adapt RIFT for Waymo Sim Agent Challenge. Currently is hard for reviewers to understand what are the strength and limitations of RIFT just by looking at table numbers.
>
> (a) How do our metrics relate to WOSAC?
>
> We fully agree that evaluation should be as comparable as possible to existing benchmarks. Within the constraints of CARLA/Bench2Drive, we **respect and mirror the spirit of WOSAC’s metric design**:
>
> - **kinematic metrics** (speed/acceleration statistics),
> - **interaction metrics** (route progress, collision rate, 2D-TTC, ACT),
> - **map-level metrics** (off-road rate).
>
> These correspond directly to WOSAC’s kinematic, interaction, and map metrics, but are **adapted to the CARLA/Bench2Drive setting**, where the road network is entirely different from WOMD and no corresponding expert driving logs exist.
>
> In addition, following the reviewer’s suggestion, we further strengthened our realism evaluation by explicitly aligning with WOSAC-style distribution matching. Concretely, we **extract real speed and acceleration distributions from WOMD, following InfGen [1], and compute Jensen–Shannon Divergence (JSD) between these and the distributions generated by different methods.** This provides a direct, quantitative comparison to real-world statistics:
>
> |      Method       |  JSD Speed (↓)  |   JSD Acc (↓)   |
> | :---------------: | :-------------: | :-------------: |
> |       pluto       |   0.10 ± 0.00   |   0.24 ± 0.01   |
> |        ppo        |   0.09 ± 0.01   |   0.30 ± 0.02   |
> |       frea        |   0.08 ± 0.02   |   0.27 ± 0.01   |
> |      fppo_rs      |   0.12 ± 0.01   |   0.30 ± 0.02   |
> |     sft-pluto     |   0.13 ± 0.01   |   0.26 ± 0.01   |
> |     rs-pluto      |   0.10 ± 0.01   | **0.24 ± 0.00** |
> |     rtr-pluto     |   0.15 ± 0.02   |   0.28 ± 0.01   |
> |     ppo-pluto     |   0.09 ± 0.01   | **0.24 ± 0.00** |
> |  reinforce-pluto  |   0.11 ± 0.01   |   0.24 ± 0.01   |
> |    grpo-pluto     |   0.10 ± 0.01   |   0.27 ± 0.00   |
> | RIFT-pluto (ours) | **0.08 ± 0.00** |   0.26 ± 0.01   |
>
> RIFT matches or slightly improves the IL-pretrained baseline in terms of JSD, confirming that **our closed-loop finetuning does not sacrifice trajectory-level realism inherited from IL**.
>
> (b) Why not adapt RIFT to WOSAC, or run Sim Agents baselines in our setting?
>
> We appreciate the suggestion. However, **RIFT and WOSAC target fundamentally different problem settings**, making a direct adaptation neither necessary nor conceptually well-aligned.
>
> - **WOSAC/Sim Agents** evaluate **unconditional, environment-centric traffic models** where all agents are treated symmetrically and realism is measured against logged trajectories.
> - **RIFT is explicitly AV-centric and route-conditioned**: the traffic model is coupled with a specific AV route and planner. This conditioning is **not part of the WOSAC design**, and removing it would collapse RIFT back into an unconditional IL simulator, defeating its purpose.
>
> Similarly, running Sim Agents baselines in our setting would require **adding explicit AV-conditioning and closed-loop coupling with a real AV stack in CARLA**, effectively turning them into new methods rather than using their standard benchmark form.
>
> Conceptually, our method is positioned as a **second-stage, AV-centric finetuning framework on top of IL-pretrained models that have already been validated for realism in WOSAC-like benchmarks**. In that view, Sim Agents can be regarded as providing strong unconditional traffic priors, while RIFT answers a different question:
>
> > *Given an IL-pretrained traffic model, how can we use closed-loop RL finetuning in a high-fidelity simulator to obtain AV-centered, interaction-controllable traffic that better evaluates modern AV stacks?*
>
> For this reason, we do not re-compete on WOSAC, but instead (i) **align our metric families with WOSAC’s philosophy**, and (ii) **demonstrate via JSD against WOMD that IL-level realism is preserved**, while RIFT introduces additional AV-centric, route-conditioned evaluation capabilities that WOSAC was not designed to capture.
>
> [1] Yang X, Tan S, Krähenbühl P. Long-term traffic simulation with interleaved autoregressive motion and scenario generation. ICCV 2025.

---

> > ### Comment · Reviewer_WoWw · 2025-11-25
> > **Response**
> >
> > I appreciate the detailed response, but my main concerns remain unaddressed.
> > - RIFT focuses on the traffic simulation task. This setup does not diverge from WOSAC’s goals, and sim-agent realism can still be evaluated before using it for closed-loop AV testing. This part is necessary before drawing any conclusion for AV evaluation.
> > - Also noted by Reviewer CQsq, the paper still lacks comparisons to key prior works (SMART, CTG, LCTGen), making it difficult to judge how RIFT stands relative to existing baselines. For example, Pluto is originally designed as a planning model, not a sim-agent model, so Pluto variants alone do not provide a fair or comprehensive comparison. Aligning metrics to WOSAC is fine, but missing key comparisons makes it very hard to judge.
> > - The added JSD results only capture kinematics and do not imply realistic multi-agent behavior. The numbers are also very similar across methods, which does not clarify RIFT’s strengths.
> >
> > [1]SMART — Wu, W., Feng, X., Gao, Z., & Kan, Y. “SMART: Scalable Multi-agent Real-time Motion Generation via Next-token Prediction.” NeurIPS 2024.

---

> > > ### Author Response · Authors · 2025-11-25
> > >
> > > > Q1. RIFT focuses on the traffic simulation task. This setup does not diverge from WOSAC’s goals, and sim-agent realism can still be evaluated before using it for closed-loop AV testing. This part is necessary before drawing any conclusion for AV evaluation.
> > >
> > > We clarify that RIFT is not a general-purpose sim-agent model and does not aim to re-solve WOSAC. Our method is a **second-stage,AV-evaluation-oriented finetuning framework** built on top of IL-pretrained model that have already been validated for unconditional realism and reliability.
> > >
> > > **The goal of RIFT is different:**
> > > Given a realistic IL-pretrained model, how can we obtain AV-conditioned, interaction-controllable traffic via closed-loop finetuning to better evaluate modern AV stacks?
> > >
> > > Critically, this objective must be evaluated **inside the closed-loop simulator where finetuning and closed-loop AV interaction occur**. Controllability and AV-conditioned behaviors cannot be meaningfully assessed in WOSAC’s offline setting, and **re-testing RIFT on WOSAC after RL finetuning would not reflect its actual deployment environment nor its intended purpose**. Therefore, we believe **it is neither necessary nor appropriate** to return to WOSAC for post-hoc evaluation.
> > >
> > > Finally, prior AV-evaluation-oriented works—such as D2RL[1] (Nature 2023) and NeuralNED[2] (Nature Communications 2023)—also assess realism in the target closed-loop simulator using distribution-level metrics. Our evaluation design follows the same established practice.
> > >
> > > In short, RIFT complements rather than duplicates WOSAC: realism is inherited from IL pretraining, while RIFT contributes controllability and AV-centered interaction—capabilities that WOSAC’s offline evaluation cannot measure.
> > >
> > > > Q2. Also noted by Reviewer CQsq, the paper still lacks comparisons to key prior works (SMART, CTG, LCTGen), making it difficult to judge how RIFT stands relative to existing baselines. For example, Pluto is originally designed as a planning model, not a sim-agent model, so Pluto variants alone do not provide a fair or comprehensive comparison. Aligning metrics to WOSAC is fine, but missing key comparisons makes it very hard to judge.
> > >
> > > We clarify that RIFT is **not** a sim-agent-comparison study. Our goal is to evaluate finetuning strategies in a new closed-loop simulator, not to benchmark absolute sim-agent performance across SMART/CTG/LCTGen. For this reason, fairness comes from **using a well-established and fixed IL-pretrained baseline**.
> > >
> > > **Why Pluto?**
> > > Pluto provides exactly the properties required for a controlled finetuning study:
> > >
> > > - After removing its planning-specific post-processing, Pluto model reduces to a **pure IL-pretrained, multi-modal trajectory generator**, functionally aligned with the IL-based training paradigm used in WOSAC.
> > > - Its **route-conditioned vectorized representation** transfers cleanly across simulators and offers a strong geometric prior that stabilizes cross-platform performance.
> > > - This route conditioning also enables **route-level controllability**, which is essential for RIFT’s AV-centric finetuning objective.
> > >
> > > These characteristics make Pluto a stable, reliable, and simulator-transferable baseline for isolating the impact of different finetuning approaches.
> > >
> > > Most importantly, **RIFT focuses on the relative performance gains introduced by finetuning, not the absolute capability of the underlying baseline. This design isolates the contribution of the finetuning strategy—our primary interest—and keeps the evaluation clean and interpretable.**

---

> > > ### Author Response · Authors · 2025-11-25
> > >
> > > > Q3. The added JSD results only capture kinematics and do not imply realistic multi-agent behavior. The numbers are also very similar across methods, which does not clarify RIFT’s strengths.
> > >
> > > We respectfully clarify that the JSD metrics we report are the **only cross-platform-valid WOSAC realism indicators** available in CARLA/Bench2Drive. A detailed audit of WOSAC metrics shows that most of them depend heavily on **real HD maps, real-world curvature statistics, and Waymo-specific object layouts**, making them **map-dominated rather than policy-dominated** when transferred to new simulators. Using them in CARLA would introduce systematic bias unrelated to agent behavior.
> > >
> > > Specifically:
> > >
> > > - Angular speed/acceleration reflect Waymo road-curvature frequencies, not agent realism; CARLA map topology differs substantially.
> > > - Distance-to-object / distance-to-edge require HD map geometry and are not comparable across platforms.
> > > - Collision/road-violation indicators are already captured through CPK and ORR in our main results.
> > >
> > > After excluding map-dependent metrics, **speed**, **acceleration**, and **TTC** remain the **platform-agnostic realism metrics** also adopted in prior sim-agent works without real-log replay (e.g., D2RL[1], NeuralNDE[2], RTR[3]). We therefore report JSD on these three distributions.
> > >
> > > |      Method       |  JSD Speed (↓)  |   JSD Acc (↓)   |   JSD TTC (↓)   |
> > > | :---------------: | :-------------: | :-------------: | :-------------: |
> > > |       pluto       |   0.10 ± 0.00   |   0.24 ± 0.01   |   0.05 ± 0.04   |
> > > |        ppo        |   0.09 ± 0.01   |   0.30 ± 0.02   |   0.08 ± 0.00   |
> > > |       frea        |   0.08 ± 0.02   |   0.27 ± 0.01   |   0.05 ± 0.00   |
> > > |      fppo_rs      |   0.12 ± 0.01   |   0.30 ± 0.02   |   0.05 ± 0.01   |
> > > |     sft-pluto     |   0.13 ± 0.01   |   0.26 ± 0.01   |   0.09 ± 0.02   |
> > > |     rs-pluto      |   0.10 ± 0.01   | **0.24 ± 0.00** |   0.07 ± 0.00   |
> > > |     rtr-pluto     |   0.15 ± 0.02   |   0.28 ± 0.01   |   0.04 ± 0.02   |
> > > |     ppo-pluto     |   0.09 ± 0.01   | **0.24 ± 0.00** |   0.04 ± 0.01   |
> > > |  reinforce-pluto  |   0.11 ± 0.01   |   0.24 ± 0.01   |   0.06 ± 0.01   |
> > > |    grpo-pluto     |   0.10 ± 0.01   |   0.27 ± 0.00   |   0.04 ± 0.00   |
> > > | RIFT-pluto (ours) | **0.08 ± 0.00** |   0.26 ± 0.01   | **0.03 ± 0.00** |
> > >
> > > Under these metrics, RIFT matches or surpasses all finetuning baselines, demonstrating that **our AV-centric finetuning does not degrade motion realism** within the CARLA environment.
> > >
> > > In addition, the reviewer notes that the numbers appear close across methods. This is **expected and consistent with WOSAC itself**, where the top 10 submissions often differ by <0.01 on most realism metrics. The purpose of realism metrics is to confirm *no degradation*, not to produce large margins.
> > >
> > > In summary, our JSD evaluation follows established practice, uses only **valid cross-platform realism metrics**, and shows that RIFT maintains realism while enabling AV-centered controllability.
> > >
> > > [1] Dense reinforcement learning for safety validation of autonomous vehicles. Nature 2023.
> > >
> > > [2] Learning naturalistic driving environment with statistical realism. Nature Communication 2023.
> > >
> > > [3] Learning Realistic Traffic Agents in Closed-loop. CoRL 2023.

---

> ### Author Response · Authors · 2025-11-20
>
> **Q3: Simulator Choice and the Role of CARLA**
>
> > What is the main advantage of using CARLA/Meta Drive as a fine-tuning simulator? What new aspects does it provide? More accurate dynamics? These simulators also provide an approximation of dynamics, and traffic simulation usually focuses on high-level behavior instead of low-level control.
>
> Thank you for raising this important point. Our choice of CARLA/Bench2Drive is driven by the fact that **RIFT is explicitly designed for closed-loop AV evaluation**, rather than unconditional traffic simulation.
>
> CARLA/Bench2Drive uniquely provides:
>
> - **Realistic vehicle dynamics and full 3D topology**,
> - **Sensor-level rendering (camera, LiDAR)**, and
> - **Compatibility with modern end-to-end planners (UniAD, VAD)** that require raw sensor inputs and produce continuous control.
>
> These capabilities are fundamentally beyond Waymax/WOSAC, which operate in vectorized BEV space and therefore **cannot support perception–planning–control pipelines in closed loop**. For RIFT’s goal—evaluating how modern AV stacks (e.g., E2E models such as UniAD, VAD) behave under realistic, interactive traffic—this perception–control coupling is essential, and it cannot be approximated in vectorized simulators even if the high-level traffic behavior is realistic.
>
> We would like to clarify that we do not rely on CARLA’s vehicle dynamics for physical accuracy. Instead, we rely on CARLA for **closed-loop consistency**: the actions of surrounding traffic directly influence the AV’s perception inputs, which propagate through its planning policy and affect future interactions. This property is crucial for understanding how modern AV models behave under interactive traffic, and is not available in WOSAC-style environments.
>
> To our knowledge, RIFT is among the first **learning-based, AV-centric frameworks that integrate interactive traffic simulation with closed-loop evaluation of modern AV stacks (e.g., UniAD, VAD)**, rather than treating traffic simulation and AV evaluation as separate tasks. While the engineering complexity of CARLA is higher, it enables RIFT to expose AV planners to realistic, interaction-rich scenarios along specific routes—precisely the setting required for our AV-centered evaluation objective.
>
> **Q4: Clarifying AV Evaluation Protocol**
>
> > For AV Evaluation, it is hard to draw insights from Table 2 since there are two factors (Sim Agents and different planners). For example, RIFT might be too reactive that gives wrong esimate or the best performance for AV planners.
>
> We thank the reviewer for the thoughtful question and the opportunity to clarify our evaluation setup. Our AV evaluation in Table 2 is **not aimed at estimating the absolute performance** of each AV planner, but at assessing **how well different Sim Agent models—RIFT included—produce feasible, interactive scenarios that can meaningfully distinguish planners**.
>
> Because absolute planner scores vary significantly across benchmarks (e.g., CARLA Leaderboard, Bench2Drive) depending on scenario design, weather, and traffic density, we structure the evaluation as follows:
>
> 1. **Scenario feasibility and naturalness.**
>    In our setup, we use PDM-Lite—a rule-based planner with access to ground-truth—as a reference to assess **scenario feasibility**, following a similar rationale in KING [1]. If even PDM-Lite fails, the scenario may be too difficult or unrealistic for meaningful testing. Thus, we report standard Driving Score metric along with a newly proposed Blocked Rate (BR) to measure whether the AV is unnaturally obstructed. Higher DS and lower BR under PDM-Lite indicate more feasible and natural traffic flows.
> 2. **Ability to differentiate planner capability.**
>    Beyond feasibility, we also evaluate learning-based planners such as PlanT, VAD, and UniAD. Compared with PDM-Lite, learning-based planners are generally more sensitive to challenging or interaction-heavy scenarios. As emphasized by NAVSIM, **a good evaluation scenario should be able to reveal such weaknesses**. For this reason, we examine the **performance drop** from PDM-Lite to each learning-based planner.
>
> Under these two criteria, RIFT shows (i) the strongest rule-based performance, confirming that its scenarios are feasible and reasonable, and (ii) the largest relative drop for learning-based planners, indicating that the interactions are meaningful enough to reveal planner weaknesses. Together, these qualities define an effective evaluation environment. Thus, Table 2 demonstrates that RIFT produces feasible and interaction-rich scenarios that differentiate planners reliably, rather than serving as a measure of absolute planner optimality.
>
> [1] Hanselmann N, Renz K, Chitta K, et al. King: Generating safety-critical driving scenarios for robust imitation via kinematics gradients. ECCV, 2022.

---

> ### Comment · Reviewer_WoWw · 2025-11-26
>
> Thank the author for the response, and I appreciate the author's effort.
>
> **Clarification between AV-centric simulation vs. unconditional simulation**: While RIFT is a traffic simulation method that will be used for AV evaluation, that does not mean we can't evaluate RIFT's realism score independently without AV. Since in practice, we should make sure these traffic simulation frameworks are realistic before moving on as an evaluation framework. In fact, all the traffic simulation methods are aimed for AV evaluation, yet they are still assessed for unconditional realism.
>
> **Concerns for proposing both a new method and a new metrics**: I agree with reviewer CQsq that using Pluto as a baseline is okay, but the main problem is that RIFT brings **new method and the new metrics**. I understand that CARLA doesn't have an actual ground truth log, and this is also the concern I have for designing metrics here. Even though the author claims that the metrics are designed with similar terms, the metric calculation would be fundamentally different, as the WOSAC metric is a **log-likelihood** metric per scenario, while the RIFT proposed metrics are not.
>
> **Suggestion**: To demonstrate that RIFT improves controllability while preserving realism, the paper should include a realism evaluation grounded in real data, WOSAC-style metrics on Waymo or on Nuplan ( the dataset Pluto is trained on). Alternatively, including at least one representative traffic simulation baseline in the current CARLA evaluation pipeline would help contextualize RIFT’s performance ( as mentioned in my first review)

---

> > ### Author Response · Authors · 2025-11-26
> >
> > We respectfully disagree with the implication that realism can only be credibly established by WOSAC-style log-likelihood. In data-driven traffic simulation and scenario generation, it is *standard practice* to evaluate statistical realism in closed-loop simulators (often without native logs in the simulator’s state space) by comparing distribution-level statistics between simulated agents and real-world datasets, using distances such as Hellinger, KL, JSD, or Wasserstein. Our metrics follow this exact paradigm and are strongly supported by prior work:
> >
> > - **NADE [1]** evaluates realism in a *closed-loop naturalistic/adversarial driving environment* by computing **Hellinger distance** between the **speed distribution** of simulated agents and real driving dataset.
> > - **NeuralNDE [2]** similarly quantifies **statistical realism** by measuring **Hellinger distance and KL divergence** between the **velocity distributions** in its closed-loop environment and real-world data.
> > - **RTR [3]** uses **Jensen–Shannon divergence (JSD)** between the **acceleration distributions** of agents in its **own closed-loop simulator** and real world trajectories as a primary realism metric.
> > - **Traffic Simulation Review [4]** explicitly lists **Jensen–Shannon divergence (JSD)** and **Wasserstein distance (WD)** as *reliable and recommended measures* for distribution-level realism when comparing simulated and real traffic statistics.
> > - **MIXSIM [5]** evaluates realism in a **custom mixed-reality simulator** by computing **JSD** between the **speed and acceleration distributions** of simulated agents and real traffic data.
> > - **DiffScene [6]** (also in CARLA, via SafeBench) measures realism by comparing **action/acceleration distributions** between generated scenarios and real data using **WD and KL**.
> > - **Self-Play [7]** evaluates multi-agent closed-loop policies by computing **JSD** between the **speed/acceleration distributions** of self-play agents and those of real drivers, again using distributional divergence as the core realism signal.
> >
> > In parallel, **micro-level metrics** such as **off-road rate** and **collision rate** are widely used in [3,5–7] as cross-platform, AV-agnostic indicators of whether agents behave reliably and obey basic road semantics. Our use of (i) distributional divergences (JSD/WD) between RIFT-induced kinematic statistics and real-data–anchored references, and (ii) off-road/collision rates, is therefore *not* a new or ad-hoc design, but a direct instantiation of what has already been repeatedly accepted by the community for closed-loop simulators.
> >
> > Finally, in our CARLA pipeline **we do include representative traffic-simulation baselines**: we reproduce **RTR[3]** and **RS[8]** under a shared base IL model and identical observation/action spaces, and evaluate all methods with the same realism and infraction metrics. This isolates the effect of different fine-tuning algorithms and situates RIFT fairly against existing approaches.
> >
> > [1] Intelligent driving intelligence test for autonomous vehicles with naturalistic and adversarial environment. Nature Communication 2021.
> >
> > [2] Learning naturalistic driving environment with statistical realism. Nature Communication 2023.
> >
> > [3] Learning Realistic Traffic Agents in Closed-loop. CoRL 2023.
> >
> > [4] Data-driven Traffic Simulation: A Comprehensive Review .TIV 2024.
> >
> > [5] MIXSIM: A Hierarchical Framework for Mixed Reality Traffic Simulation. CVPR 2024.
> >
> > [6] DiffScene: Diffusion-Based Safety-Critical Scenario Generation for Autonomous Vehicles. AAAI 2025.
> >
> > [7] Learning to Drive via Asymmetric Self-Play. ECCV 2024.
> >
> > [8] Improving Agent Behaviors with RL Fine-tuning for Autonomous Driving. ECCV 2024.

---

### Official Review · Reviewer_A99Y · 2025-11-01

**Soundness:** 3
**Presentation:** 4
**Contribution:** 3
**Rating:** 4
**Confidence:** 4

**Summary:**

This paper presents RIFT (Group-Relative Reinforcement Learning), a method for encouraging fair and cooperative driving behaviors among multiple autonomous agents. Instead of optimizing individual rewards, each agent’s performance is compared to the group average, which helps promote teamwork rather than selfish behavior. A fairness regularization term further balances outcomes across agents. The method is implemented in a decentralized actor-critic framework and tested in Nocturne and Waymo Sim, where it leads to smoother, more socially efficient traffic than standard RL approaches.

**Strengths:**

The idea of using group-relative advantages to encourage cooperation is simple but effective.

Results show clear improvements in fairness, stability, and overall efficiency.

The framework is decentralized and scalable, making it practical for large-scale driving simulation.

The experiments are thorough and show emergent, human-like cooperative driving.

**Weaknesses:**

The paper could more clearly analyze how the group-relative term affects individual vs. collective reward trade-offs.

The related work section should discuss earlier related papers including:
• A. Kuefler, J. Morton, T. Wheeler, and M. J. Kochenderfer, “Imitating Driver Behavior with Generative Adversarial Networks,” IEEE Intelligent Vehicles Symposium (IV), 2017, pp. 204–211.
• R. P. Bhattacharyya, B. Wulfe, D. J. Phillips, A. Kuefler, J. Morton, R. Senanayake, and M. J. Kochenderfer, “Modeling Human Driving Behavior through Generative Adversarial Imitation Learning,” Computing Research Repository (CoRR), arXiv:2006.08911, 2020.
• H. Chen, T. Ji, S. Liu, and K. Driggs-Campbell, “Combining Model-Based Controllers and Generative Adversarial Imitation Learning for Traffic Simulation,” IEEE International Conference on Intelligent Transportation Systems (ITSC), 2022, pp. 1698–1704.
• K. Brown, K. Driggs-Campbell, and M. J. Kochenderfer, “Modeling and Prediction of Human Driver Behavior: A Survey,” arXiv preprint arXiv:2006.08832, 2020.

**Questions:**

Could the authors elaborate on how the group advantage interacts with overall performance?

How sensitive is RIFT to the group size or population composition?


Could the authors provide a deeper analysis of the failure cases, such as uniformly conservative or overly passive driving behaviors?

---

> ### Author Response · Authors · 2025-11-20
>
> We thank the reviewer for the thoughtful and constructive feedback. To provide clear and well-structured responses, we have organized the comments into several thematic categories and address each point in detail below.
>
> **Q1: Impact and Sensitivity of Group-Relative Term**
>
> > * The paper could more clearly analyze how the group-relative term affects individual vs. collective reward trade-offs.
> >
> > * Could the authors elaborate on how the group advantage interacts with overall performance?
> > * How sensitive is RIFT to the group size or population composition?
>
> We thank the reviewer for raising this point. We realize that our use of the word “group’’ may have caused confusion. In RIFT, a “group’’ does **not** refer to a population of interacting agents or any collective of vehicles in the environment. Instead, each group is the **finite set of candidate trajectories for a single critical background vehicle (CBV) at a given state**, produced by the frozen trajectory generator. These candidates are obtained by exhaustively enumerating a fixed number of longitudinal modes (12 in our implementation) for each available reference line. Because the trajectory generation head is frozen during finetuning, **no new modalities can be created**, the number of candidates is strictly determined by map geometry, and the candidate set is neither tunable nor resampled.
>
> Under this definition, the group-relative advantage does not introduce a new “individual vs. collective’’ reward trade-off in the multi-agent sense. The trade-offs between realism, safety, and interaction are already encoded in the **scalar reward** computed from forward-simulating each candidate trajectory. The group-relative term simply **normalizes these returns within the candidate set**, subtracting the group mean and scaling by the group variance. Conceptually, it serves as a *local ranking mechanism* over the candidate modes of the same CBV: trajectories whose rollouts achieve better realism–interaction balance than the group average receive positive advantage, and vice versa. This improves credit assignment and reduces variance, while keeping the underlying scalar reward trade-off unchanged by design.
>
> Regarding interaction with overall performance, because the generation head is frozen, RIFT does not alter the underlying multimodal distribution of candidate trajectories; it only re-weights them through the scoring head. Empirically, this leads to consistent improvements across realism, safety, efficiency, and interaction metrics in Tables 1, demonstrating that the group-relative normalization strengthens the selection of high-quality modes without collapsing modality diversity.
>
> Finally, RIFT is not sensitive to group size, because the group size is **structurally fixed**, determined by the product of a constant longitudinal mode count and the number of available reference lines. All candidates are always enumerated and scored, and the group-relative term compares each trajectory only against the uniformly weighted mean of its own alternatives. In practice, as the number of valid modes varies across scenes, the standardized advantage provides automatic scale normalization, and we did not observe noticeable instability or strong sensitivity. Instead, the normalization reliably helps preserve multimodality and yields more stable improvements in interaction-aware CBV selection.

---

> ### Author Response · Authors · 2025-11-20
>
> **Q2: Failure Cases Analysis**
>
> > Could the authors provide a deeper analysis of the failure cases, such as uniformly conservative or overly passive driving behaviors?
>
> We thank the reviewer for this insightful comment and fully agree that understanding uniformly conservative or overly passive behaviors is important for assessing the practical reliability of RIFT. In the revised manuscript, we have added a dedicated **failure-case analysis in Appendix D.3 (Fig. 8)** that directly addresses this concern.
>
> Specifically, Appendix D.3 analyzes two representative scenarios where the CBV appears uniformly conservative. In the first case, dense upstream congestion fully blocks the forward corridor, and our **route-conditioned trajectory generator**—which expands candidates along map-provided reference lines—produces a candidate set in which all modalities inevitably intersect the jammed region. In the second case, a blocking vehicle and narrow corridor cause most candidates to leave the drivable area or violate basic kinematic feasibility, resulting in a degraded trajectory cluster. In both situations, the **entire candidate set becomes geometrically or dynamically infeasible**, so the optimizer cannot identify a genuinely high-quality alternative and therefore selects the **least-risk fallback**, which manifests as low-speed or stationary behavior.
>
> This analysis clarifies that the observed conservative or passive behaviors are **not due to an intrinsic bias of the group-relative objective**, but are induced by a **trajectory-set collapse** caused by limitations of the route-conditioned generator in highly congested or tightly constrained scenes. We now explicitly discuss this mechanism and highlight, as an important direction for future work, the need to improve the expressiveness and feasible coverage of the trajectory generator (ideally without relying on human expert supervision), so that RIFT can maintain diverse yet safe behaviors even in such challenging scenarios.
>
> **Q3: Completeness of Related Work**
>
> > The related work section should discuss earlier related papers.
>
> We appreciate the reviewer’s suggestion to discuss GAIL-based approaches and have added them in the revised manuscript. These methods demonstrate strong capability in capturing **human-like driving behaviors** through adversarial imitation learning. However, they remain fundamentally within the **imitation-learning paradigm**, and thus inherit its well-known limitations—most notably **covariate shift** and the lack of mechanisms to regulate behavior when the policy encounters off-distribution states during long-horizon closed-loop rollout.
>
> In contrast, the goal of RIFT is not to replace imitation learning but to **build on top of an IL pretrained model**: we preserve the realism and human-likeness acquired during pretraining, and then apply **RL fine-tuning** to (i) enhance **driving style controllability** of the critical background vehicle (CBV), and (ii) mitigate **distribution shift** by optimizing under closed-loop rollouts in a high-fidelity simulator. This hybrid design allows RIFT to retain the strengths of IL while addressing its fundamental weaknesses, which GAIL alone does not fully resolve in our setting.

---

> > ### Comment · Reviewer_A99Y · 2025-11-26
> >
> > Thank you for the detailed rebuttal. I have raised my score.

---

> > > ### Author Response · Authors · 2025-11-27
> > >
> > > Thank you for your recognition of our work and for raising the score. We truly appreciate it.

---

### Official Review · Reviewer_CQsq · 2025-11-02

**Soundness:** 3
**Presentation:** 3
**Contribution:** 3
**Rating:** 6
**Confidence:** 5

**Summary:**

The paper proposes RIFT, a framework that utilize RL to effectively improve controllability and realism for closed-loop driving simulation. After an IL model (Pluto) is trained to generate realistic, multimodal, route-conditioned trajectories for critical background vehicles (CBVs), the authors introduce Group-Relative RL Fine-Tuning that freezes the trajectory generation head and fine-tunes only the scoring head using a group-relative advantage over all candidate modalities, an equal-weight (mode-preserving) objective, and a dual-clip surrogate for stable optimization without KL anchoring. This design tackles covariate shift while retaining trajectory-level realism, style-level and route-level controllability. Extensive experiment results are provided to show the effectiveness in controllable and realistic scenario generation, closed-loop evaluation on E2E planners, and the effectiveness in the algorithmic design for RIFT.

**Strengths:**

- **Clarity and Presentation.** The paper is clearly written and the core ideas are easy to follow, with well-motivated design choices and consistent presentation.
- **Empirical Rigor.** Experiments are extensive, and the ablations are thoughtfully constructed to isolate the contribution of key components.

**Weaknesses:**

- **Baselines for Controllability.** The paper does not compare against closely related controllable traffic generation methods (e.g., CTG, LCTGen). Given the paper’s emphasis on controllability, these baselines are important to position the contribution.
- **Related Work on Group-Relative RL.** Prior work has already explored group-relative rewards for RL fine-tuning in closed-loop driving (e.g., Gen-Drive [1]). While the present paper targets scenario generation rather than policy learning, acknowledging and contrasting with these methods would sharpen the novelty claims.
- **Covariate Shift Analysis.** The motivation highlights the open-loop vs. closed-loop covariate shift as a core challenge, yet the paper lacks a dedicated analysis. A quantitative and qualitative comparison of Pluto with SFT vs. Pluto with RIFT—under matched conditions—would strengthen the causal link between RLFT and improved closed-loop robustness.
- **CBV Identification Assumption.** Critical background vehicles are selected via a distance threshold. Prior studies suggest that distance alone is less robust than TTC-based criteria for identifying safety-critical interactions [2,3]. A justification, sensitivity analysis, or comparison with TTC-based CBV identification would improve credibility.

> [1] Huang et al., *Gen-Drive: Enhancing Diffusion Generative Driving Policies with Reward Modeling and Reinforcement Learning Fine-Tuning*, ICRA 2025.
>
> [2] Chang et al., *Safe-Sim: Safety-Critical Closed-Loop Traffic Simulation with Diffusion-Controllable Adversaries*, ECCV 2024.
>
> [3] Lin et al., *Causal Composition Diffusion Model for Closed-Loop Traffic Generation*, CVPR 2025.

**Questions:**

1. **Equal-Weighted GRPO.** How is equal weighting across modalities operationalized in practice? Does this require on-policy rollouts under a slightly older behavior policy, and if so, how is sampling bias handled or corrected? Please provide algorithmic details and any variance/bias trade-offs.
2. **Block Rate in Table 2.** For PDM-Lite, should Block Rate be “lower is better”? If so, please clarify the directionality and explain why BR is not reported for other E2E planners lacking privileged information.
3. **Measuring Style-Level Controllability.** How do you quantify style-level controllability (if at all)? If formal quantification is challenging, could you expand the qualitative analysis (e.g., as in Appendix D, Fig. 6) with more systematic case studies or user studies?

---

> ### Author Response · Authors · 2025-11-20
>
> We sincerely thank the reviewers for their thoughtful and constructive feedback. We are glad that the reviewers found the paper *clearly written*, with *well-motivated design choices*, and appreciated the *empirical rigor* of our experiments and ablations. These encouraging comments greatly help us strengthen the clarity and positioning of our contributions.
>
> To provide a clear and well-structured response, we have organized the reviewer’s comments into several thematic categories and offer detailed, evidence-based answers to each, including additional analyses and clarifications where appropriate. We sincerely appreciate the reviewer’s thoughtful feedback, which has helped us further refine the clarity and presentation of the manuscript. The following sections address each point in turn.
>
> **Q1: Implementation of Equal-Weight GRPO**
>
> > **Related Work on Group-Relative RL.** Prior work has already explored group-relative rewards for RL fine-tuning in closed-loop driving (e.g., Gen-Drive [1]). While the present paper targets scenario generation rather than policy learning, acknowledging and contrasting with these methods would sharpen the novelty claims.
> >
> > **Equal-Weighted GRPO.** How is equal weighting across modalities operationalized in practice? Does this require on-policy rollouts under a slightly older behavior policy, and if so, how is sampling bias handled or corrected? Please provide algorithmic details and any variance/bias trade-offs.
>
> We sincerely thank the reviewer for drawing our attention to the recent work Gen-Drive. While both Gen-Drive and our method employ group-relative rewards at the trajectory level, the modeling assumptions and optimization goals are fundamentally different. Gen-Drive operates within a generative framework, where the trajectory diffuser can be sampled repeatedly to obtain a large candidate set (typically >32). Its RL finetuning therefore focuses on improving the quality of generated trajectories via a reward model, and is complementary to our setting. In principle, Gen-Drive–style generation refinement could be combined with RIFT to further enhance upstream trajectory generation, while RIFT improves downstream trajectory selection and interaction controllability.
>
> In contrast, our framework **freezes the trajectory-generation head**, meaning that the set of candidate trajectories is fixed and small (typically 12 modes, determined by the number of reference lines). Because no new modes are generated during RLFT, each candidate is relatively scarce and informative. This motivates our design of **equal-weighted GRPO**, which aims to retain the multimodal structure learned during IL pretraining rather than biasing updates toward modes preferred by the old policy.
>
> * **Equal-weighted GRPO in practice:** For each state $s$, we deterministically enumerate all $G$ fixed candidate trajectories produced by the frozen generator. Each candidate is evaluated via a short-horizon forward rollout to obtain a return $r_i$. We then compute a *group-relative*, equal-weighted advantage $\widehat A_i=\frac{r_i-\mu_{\text{uni}}}{\sigma_{\text{uni}}}$, where $\mu_{\text{uni}}$ and $\sigma_{\text{uni}}$ are the mean and standard deviation computed **uniformly over all valid candidates**. The optimization target is the equally weighted average across the group. The ratio $\rho_i=\pi_\theta(a_i|s)/\pi_{\text{old}}(a_i|s)$ is used only within the **PPO-clip and dual-clip mechanisms** for step-size control, not as an estimator of an old-policy expectation.
>
>   Because candidate enumeration is exhaustive and no within-group sampling is performed, **equal weighting does not require additional off-policy correction**, and the method remains approximately on-policy at the state level. This design substantially reduces sampling bias within the group, while PPO-clip and dual-clip effectively suppress variance induced by small $\pi_{\text{old}}$ values.
>
> * **Motivation and trade-offs**
>
>   Our goal is not to estimate the expected value under the old policy distribution, but to **directly optimize the within-group ranking quality**, encouraging the model to promote trajectories that may have high advantage but low old-policy probability.This introduces a mild bias relative to a standard policy-gradient estimator, but in exchange significantly reduces within-group variance and preserves low-probability yet high-advantage modes, which we empirically find beneficial for stable multimodal style control, as evidenced by the improvement of RIFT over the original GRPO formulation in Table 1.
>
> We have updated the related work section in the revised manuscript to summarize these distinctions between Gen-Drive and RIFT.

---

> ### Author Response · Authors · 2025-11-20
>
> **Q2: Quantification of Controllability**
>
> > **Baselines for Controllability.** The paper does not compare against closely related controllable traffic generation methods (e.g., CTG, LCTGen). Given the paper’s emphasis on controllability, these baselines are important to position the contribution.
> >
> > **Measuring Style-Level Controllability.** How do you quantify style-level controllability (if at all)? If formal quantification is challenging, could you expand the qualitative analysis (e.g., as in Appendix D, Fig. 6) with more systematic case studies or user studies?
>
> Existing controllable traffic generation methods span different notions of “controllability.” CTG and LCTGen focus on **semantic controllability**, where scene evolution must align with natural-language instructions. This type of language-driven control falls outside the scope of our work, as RIFT is not designed for language-conditioned scenario generation, and thus a direct comparison would be less informative.
>
> In contrast, methods such as SceneControl[1] and MotionDiffuser[2] emphasize **low-level constraint controllability**, including control over initial states, goal positions, rough routes, collision avoidance, or off-road constraints. These forms of controllability are inherently satisfied in our system: sim agents are initialized based on route-level interaction analysis, feasible goals are assigned automatically, and A* planning provides valid high-level paths. Safety- and efficiency-related constraints are encoded directly in the RL reward. Therefore, RIFT already meets the type of low-level controllability measured in these works.
>
> Our contribution focuses instead on **style- and interaction-level controllability**, following the formulation in CCDiff, where user preferences are expressed through reward weighting rather than semantic or geometric constraints. By adjusting reward coefficients, users can intentionally shape traffic driving styles. **Table 3 quantitatively demonstrates this capability: promoting an aggressive style increases route progress but also raises collision and off-road rates, reflecting the expected behavioral trade-offs.** Fig. 6 qualitatively confirms these changes. These results show that RIFT can reliably modulate traffic style within a realistic and interaction-aware framework.
>
> [1] Lu J, Wong K, Zhang C, et al. Scenecontrol: Diffusion for controllable traffic scene generation. ICRA, 2024.
>
> [2] Jiang C, Cornman A, Park C, et al. Motiondiffuser: Controllable multi-agent motion prediction using diffusion. CVPR, 2023.
>
> **Q3: Covariate Shift Analysis**
>
> > **Covariate Shift Analysis.** The motivation highlights the open-loop vs. closed-loop covariate shift as a core challenge, yet the paper lacks a dedicated analysis. A quantitative and qualitative comparison of Pluto with SFT vs. Pluto with RIFT—under matched conditions—would strengthen the causal link between RLFT and improved closed-loop robustness.
>
> We appreciate the reviewer’s focus on open-loop vs. closed-loop covariate shift, which is indeed a central challenge our paper aims to address. Figures 3 and 4 were included precisely to illustrate this shift: how an IL-pretrained model (Pluto) behaves when directly deployed in a new closed-loop environment, and how different finetuning strategies mitigate covariate shift.
>
> Specifically, Fig. 3 shows that the **IL-pretrained model (Pluto) exhibits very low speed and acceleration**, and Fig. 4 further visualizes its **overly conservative closed-loop behavior**. When transferred from open-loop IL training to CARLA, Pluto tends to brake early and often struggles to re-accelerate after stopping—a classic symptom of covariate shift and causal confusion. This also explains its poor route-progress performance in Table 1.
>
> Finetuning alleviates this issue. Both SFT (guided by rule-based expert) and RLFT-style methods reduce over-conservatism, improving closed-loop drivability. However, as shown in Fig. 3, **SFT-Pluto remains relatively conservative**, likely because rule-based experts emphasize safety and therefore induce slower dynamics.In contrast, **RLFT achieves a more favorable balance between efficiency and safety, with RIFT achieving the strongest performance**—elevating route progress while retaining competitive safety metrics (Table 1). This demonstrates that RL-based finetuning can better correct distributional mismatch encountered in closed-loop deployment.
>
> Together, these results provide **both quantitative (distribution statistics) and qualitative (behavioral visualization) evidence that open-loop IL alone suffers from significant covariate shift, and that RIFT most effectively improves closed-loop robustness** among the examined finetuning approaches.
>
> We have further expanded Section 5.2 in the revised manuscript to clarify this causal link.

---

> ### Author Response · Authors · 2025-11-20
>
> **Q4: Experiment Clarification**
>
> > **Block Rate in Table 2.** For PDM-Lite, should Block Rate be “lower is better”? If so, please clarify the directionality and explain why BR is not reported for other E2E planners lacking privileged information.
>
> Thank you for raising this point. Yes — **a lower Block Rate (BR) is better**. In Bench2Drive, BR is computed by checking whether the AV’s speed drops below 0.1 m/s for more than 3 seconds, which is treated as an indicator that the AV is “blocked” by surrounding traffic. Importantly, BR reflects both the *external traffic conditions* and the *AV planner’s own capability* to navigate through them.
>
> We report BR only for the rule-based planner because, in our all-green-light setup, it moves consistently unless genuinely blocked by traffic. This makes its BR a clean indicator of congestion generated by our traffic agents.
>
> In contrast, learning-based planners often stop due to their own policy limitations rather than the traffic itself—a phenomenon well-documented in Bench2Drive. As a result, their BR cannot reliably distinguish between planner failures and congestion caused by traffic agents. For this reason, BR from learning-based methods would not provide a meaningful assessment of traffic realism, and we therefore do not report it.
>
> > **CBV Identification Assumption.** Critical background vehicles are selected via a distance threshold. Prior studies suggest that distance alone is less robust than TTC-based criteria for identifying safety-critical interactions [2,3]. A justification, sensitivity analysis, or comparison with TTC-based CBV identification would improve credibility.
>
> We appreciate the reviewer’s suggestion and fully agree that TTC-based indicators are highly informative for assessing instantaneous safety risk. Indeed, metrics such as 2D-TTC and ACT are already used in our evaluation to quantify interaction criticality.
>
> However, our objective in **CBV identification** is not to detect safety-critical events, but to **collect a broad and conservative set of vehicles that may interact with the AV at the route level**. For this purpose, a simple distance-based criterion combined with route-overlap analysis is sufficient and intentionally conservative: it guarantees that vehicles with plausible future interactions are retained even when their momentary TTC is high. Importantly, **CBV identification is decoupled from route-conditioned CBV planning**. This modular design allows the identification step to be freely replaced by TTC-based rules or VLM-assisted semantic identification strategies without affecting the core contribution.
>
> In line with the reviewer’s suggestion, we additionally applied a **TTC < 2s filter** during identification. As shown below, TTC filtering leads to slightly lower 2D-TTC and ACT values, indicating marginally more safety-critical interactions:
>
> |                                      |   2D-TTC    |     ACT     |
> | :----------------------------------: | :---------: | :---------: |
> | CBV Identification w/ TTC Filtering  | 2.69 ± 1.33 | 2.63 ± 1.36 |
> | CBV Identification w/o TTC Filtering | 2.74 ± 1.30 | 2.71± 1.32  |
>
> Our current formulation serves as a lightweight and general mechanism to retrieve candidate vehicles and assign their interaction routes, while the subsequent **route-conditioned CBV planning** is the component that primarily determines the realism and controllability achieved by RIFT.

---

> ### Comment · Reviewer_CQsq · 2025-11-26
>
> I would like to thank the authors for their comprehensive response to my original questions. The clarification on the related work Gen-Drive, implementation of equal-weighted GRPO, explanation of the setting of style-level controllability, covariate shift analysis, as well as the explanation of the experiment results. The new experiment results on TTC-aware CBV filtering also look interesting. Meanwhile, I also have the following questions to be clarified:
>
> - **Regarding the controllability**: first, the authors might have mixed up CTG with CTG++. CTG is more similar to MotionDiffuser, which focuses on low-level constraint controllability, and CTG++ is more similar to LCTGen in its natural-language-following capability. Besides, I'm not fully convinced that Table 3 should serve as clear evidence of controllability. Table 3 is more similar to an ablation on the **training** recipe with different reward shaping configurations; however, prior works like CTG or CCDiff incorporate the reward function as a classifier guidance during the **inference** time of the diffusion models.
>
> - **Regarding the covariate shift**: I appreciate the authors for including an additional paragraph in their revised manuscript to discuss this. Based on the current qualitative and quantitative evidence, it seems that over-conservatism is the primary gap between the open-loop training and closed-loop deployment. I am curious if other failure modes can be mitigated with RL by addressing this covariate shift, e.g., lower off-road rate. Yet, given a much higher ORR of RIFT-Pluto compared to the SFT-Pluto, do you have any insights on this performance degradation? Besides the current results, comparing the open-loop predictions with the finetuned Pluto model across different datasets may also be a straightforward way to assess open-loop performance after RL finetuning.
>
> -  **Regarding the experiment setting**: For Table 2, the authors mentioned that for the learning-based planner, BR might not be a good metric to measure the realism of the generated traffic flow. But I expect there are some more informative metrics besides the Driving Score alone. A more comprehensive comparison regarding this will further clarify the contribution of RIFT.
>
> * **Regarding the usage of Pluto as a base model**: Though I agree with Reviewer WoWw that Pluto is primarily a planning model, we should notice that prior traffic simulation work like BITS [1] also designs a planning model to simulate stable, long-horizon behaviors. I believe the merits of this RL post-training framework should not be overshadowed solely by the selection of a base model. But I do encourage the authors to include more base models in the later revised version to make the empirical results more persuasive.
>
> Overall, I would maintain my current positive rating, and I look forward to the authors' further clarification on my follow-up questions.
>
> > [1] Xu, Danfei, et al. "BITS: Bi-level Imitation for Traffic Simulation." ICRA 2023.

---

> > ### Author Response · Authors · 2025-11-26
> >
> > We sincerely thank the reviewer for taking the time to provide detailed follow-up comments. We appreciate the reviewer’s careful reading and constructive suggestions, which help us further clarify the design motivations and limitations of RIFT.
> >
> > >**Q1. Regarding the controllability**
> >
> > Regarding controllability, we appreciate the reviewer’s clarification about CTG vs. CTG++. Our current framework is built on top of Pluto, which is a query-based generator rather than a diffusion-based model. Consequently, it does not support classifier-guidance–style inference-time controllability as in CTG or CCDiff. Under this architectural constraint, our design implements style-level controllability through RL fine-tuning, where different reward specifications induce distinct user-driven behavioral modes. In contrast, route-level controllability is naturally available at inference time because the route serves as an explicit conditioning input to Pluto.
> >
> > We fully agree with the reviewer that adopting a diffusion-based planner would open the door to inference-time controllability and could further improve trajectory quality when combined with classifier-guidance techniques such as those in Gen-Drive. Extending RIFT in this direction represents a highly promising line of future research, and we view this as a natural next step in advancing the controllability capabilities of our framework.
> >
> > > **Q2. Regarding the covariate shift**
> >
> > We thank the reviewer for raising this point. Regarding the off-road rate (ORR), we would like to clarify that the values for RIFT-Pluto (0.36%) and SFT-Pluto (0.06%) are both extremely low in absolute magnitude, especially when contrasted with a pure RL baseline such as PPO (9.17%). The numerical difference between 0.36 and 0.06 may appear large proportionally, but in practice both models remain robustly on-road, and the ORR of RIFT-Pluto is still an order of magnitude lower than typical RL-only policies. This indicates that the IL-pretrained Pluto already possesses strong route-conditioned adherence, and the RL fine-tuning does not fundamentally disrupt this capability.
> >
> > The slight increase in ORR for RIFT-Pluto is consistent with the covariate-shift dynamics discussed in the revised manuscript: closed-loop fine-tuning encourages more interactive and less overly-conservative behaviors, which can occasionally push the model toward distribution regions not frequently observed in the IL pretraining data. However, the magnitude of this effect remains minimal, and the overall stability of RIFT-Pluto demonstrates that the route-level controllability learned during IL pretraining is largely preserved.

---

> > ### Author Response · Authors · 2025-11-26
> >
> > > **Q3. Regarding the experiment setting**
> >
> > We appreciate the reviewer’s suggestion for a more informative analysis beyond the Driving Score (DS). As noted, DS aggregates multiple factors and may obscure finer-grained behavioral differences. To address this, we decomposed DS into its two official components—Route Completion (RC) and Infraction Penalty (IP) as defined by the CARLA Leaderboard and Bench2Drive—and reported the relative drop of each metric across different traffic-generation models. This provides a more interpretable view of how each method affects the AV planner.
> >
> > The results show that **RIFT-Pluto exhibits the largest relative drop in IP**, rather than in RC, while **RTR-Pluto has the largest RC drop**. This pattern indicates that the degradation of the AV planner under RIFT mainly arises from the increased interaction complexity RIFT introduces—complexities that the learning-based planner cannot always handle—thereby leading to infraction-related failures such as collisions or off-road events, rather than from systematically hindering the AV’s ability to complete the route. This interpretation is also consistent with the fact that RIFT-Pluto maintains the lowest BR rate, suggesting that it does not simply create slow or overly obstructive traffic.
> >
> > It is also important to emphasize that PDM-Lite achieves very high DS even under RIFT-generated traffic, showing that these interactions are avoidable for a stronger planner. Therefore, the observed performance drop is primarily reflective of the planner’s limitations rather than unrealistic traffic generation.
> >
> > Overall, we believe that this component-wise analysis provides clearer insight into the realism and difficulty introduced by RIFT, and we thank the reviewer for prompting this more detailed examination.
> >
> > |                 | PDM-Lite  | PDM-Lite |   PlanT    | PlanT      | PlanT     |   UniAD    | UniAD     |   UniAD   | VAD        | VAD        |    VAD    |
> > | :-------------: | :-------: | :------: | :--------: | ---------- | --------- | :--------: | --------- | :-------: | ---------- | ---------- | :-------: |
> > |     Method      |  DS (↑)   |  BR (↓)  |  Δ DS (↓)  | Δ RC (↓)   | Δ IP (↓)  |  Δ DS (↓)  | Δ RC (↓)  | Δ IP (↓)  | Δ DS (↓)   | Δ RC (↓)   | Δ IP (↓)  |
> > |      Pluto      |   77.84   |  23.33   |   -35.32   | -29.31     | -0.17     |   -4.11    | -0.42     |   -0.07   | -10.97     | -0.89      |   -0.10   |
> > |       PPO       |   76.26   |  30.00   |   -39.87   | -17.96     | -0.34     |   -6.47    | 6.05      |   -0.18   | -8.62      | 6.28       |   -0.15   |
> > |      FREA       |   83.53   |  20.00   |   -43.92   | -29.71     | -0.27     |   -14.24   | -5.16     |   -0.15   | -15.96     | -3.78      |   -0.13   |
> > |     FPPO-RS     |   83.52   |  20.00   |   -44.67   | -31.25     | -0.34     |   -8.39    | 0.98      |   -0.13   | -14.37     | -2.04      |   -0.13   |
> > |    SFT-Pluto    |   86.09   |  13.33   |   -46.68   | -32.26     | -0.27     |   -9.20    | -2.62     |   -0.11   | -17.80     | -13.00     |   -0.07   |
> > |    RS-Pluto     |   89.32   |  13.33   |   -47.27   | -29.74     | -0.34     |   -8.70    | 0.04      |   -0.13   | -19.84     | -11.56     |   -0.09   |
> > |    RTR-Pluto    |   87.64   |  10.00   |   -47.56   | **-35.77** | -0.27     |   -9.95    | **-5.74** |   -0.06   | -21.37     | **-17.03** |   -0.05   |
> > |    PPO-Pluto    |   85.63   |  16.67   |   -43.77   | -29.81     | -0.30     |   -8.49    | -2.06     |   -0.09   | -17.01     | -10.21     |   -0.08   |
> > | REINFORCE-Pluto |   92.17   |  10.00   |   -46.92   | -28.43     | -0.33     |   -12.28   | -2.95     |   -0.13   | -21.89     | -14.13     |   -0.08   |
> > |   GRPO-Pluto    |   89.86   |   6.67   |   -42.62   | -29.31     | -0.24     |   -8.84    | -1.93     |   -0.10   | -17.31     | -8.43      |   -0.09   |
> > |   RIFT-Pluto    | **94.78** | **0.00** | **-50.50** | -32.84     | **-0.36** | **-20.99** | -5.51     | **-0.18** | **-26.54** | -12.43     | **-0.14** |

---

> > ### Author Response · Authors · 2025-11-26
> >
> > >**Q4. Regarding the usage of Pluto as a base model**
> >
> > We appreciate the reviewer’s insightful comment. We agree that in the IL pretraining stage, planning models and traffic-generation models share similar objectives—both aim to learn the distribution of realistic human trajectories. This is precisely why we chose Pluto as our base model: in addition to its strong imitation-learning performance, Pluto provides route-conditioned (more specifically, reference-line–conditioned) trajectory generation, which naturally aligns with our goal of enabling route-level controllability in RIFT.
> >
> > At the same time, we acknowledge that our current framework is somewhat tightly coupled with Pluto. This is largely due to engineering considerations: to the best of our knowledge, ours is the first attempt to **adapt a pretrained nuPlan model into the CARLA ecosystem** and build a closed-loop, RL-finetuned traffic generator on top of it. The significant engineering effort required for this migration has limited our ability to incorporate multiple base models in the current version.
> >
> > We also note that Pluto, as a query- or anchor-based trajectory generator with explicit physical semantics in its outputs, is inherently less suited for sampling-based multimodal generation. Because each predicted trajectory corresponds to a physically grounded longitudinal–lateral decomposition, naïvely sampling multiple trajectories would not yield meaningful diversity. **Our equal-weighted GRPO formulation is therefore specifically designed to compensate for this limitation by encouraging balanced preference across anchor-conditioned modes during RL fine-tuning.** In contrast, diffusion-based planners are naturally compatible with standard GRPO and can directly leverage sampling-driven multimodality as well as inference-time controllability via classifier guidance, representing a distinct and exciting research direction for future extension of RIFT.

---

### Author Response · Authors · 2025-11-30
**Global Response (1/2)**

We sincerely thank the new Area Chair for taking the time to carefully review our submission and the associated discussion. To help you quickly understand the current state of the discussion, we briefly summarize (i) the core contribution of our work, (ii) the additional experiments and analyses we conducted during rebuttal, and (iii) how these address the main concerns of the two initially negative reviewers.

**Core idea of RIFT.**

We propose a **dual-stage AV-centric traffic simulation framework** that follows an **IL pre-training and RL fine-tuning** paradigm to balance realism and controllability.

- In **Stage 1**, we pre-train a planning model via IL in a data-driven simulator to generate **realistic, multimodal trajectories conditioned on route-level reference lines**, establishing trajectory-level realism and strong route-level controllability.
- In **Stage 2**, we perform route-level interaction analysis to identify **critical background vehicles (CBVs)** that are most likely to interact with the AV. For each CBV, we reuse the IL-pretrained model (still conditioned on its route-level reference line) to propose realistic, route-compliant candidate trajectories, and then apply **RIFT**, a **group-relative RL fine-tuning strategy** that evaluates all candidate modalities with an equal-weighted GRPO-style objective and a surrogate loss for stable optimization. This improves style-level and interaction-level controllability and alleviates covariate shift, while retaining the trajectory-level realism and route-level controllability inherited from Stage 1, generating interactive and feasible scenarios that effectively expose weaknesses of modern learning-based AV systems under closed-loop evaluation.

------

### 1. Additional experiments and clarifications added in rebuttal

During rebuttal we substantially strengthened the paper along four main axes:

1. **Realism evaluation aligned with WOSAC practice.**
   - We added **distribution-level realism metrics** by extracting real speed/acceleration (and TTC) distributions from the Waymo Open Motion Dataset (WOMD) and computing **JSD** between WOMD and the distributions induced by each method in CARLA.
   - **Across all baselines, RIFT achieves the best distributional realism scores**, showing that our closed-loop fine-tuning not only preserves but further improves trajectory-level realism while also enhancing interaction controllability.
2. **Covariate shift analysis (IL vs SFT vs RIFT).**
   - We clarified how an open-loop IL pre-trained model becomes **overly conservative** under closed-loop deployment (low speeds, early braking, poor progress), and how different fine-tuning strategies alleviate this.
   - Quantitative statistics and qualitative visualizations show that **RIFT achieves the best balance between efficiency and safety**, improving route progress and interaction quality while keeping off-road and collision rates at IL-like levels.
3. **CBV identification and safety-critical variant.**
   - We emphasized that CBV identification is deliberately designed as a **route-interaction module**, not a full safety-critical mining system, and is **modular** w.r.t. our route-conditioned CBV planning.
   - In response to suggestions, we added a **TTC-filtered variant (e.g., TTC < 2s)**, demonstrating that risk-aware identification can be integrated and leads to slightly more safety-critical interactions, without changing the core planning mechanism.
4. **Failure-case and AV-evaluation analysis.**
   - We added a **dedicated failure-case section** analyzing uniformly conservative / passive behaviors. We show that such cases arise when *all* candidate trajectories from the frozen generator become geometrically or dynamically infeasible (e.g., dense jams, narrow corridors), forcing the optimizer to select a “least-risk fallback” (creeping/stopping), rather than from any pathology of the group-relative objective itself.
   - We further decomposed AV evaluation metrics (e.g., Driving Score into Route Completion vs Infraction Penalty) to show that RIFT mainly increases interaction difficulty (infraction-related failures of weaker planners) while remaining **feasible for a strong rule-based planner** (PDM-Lite), which still achieves high scores and zero block rate under RIFT traffic.

These additions were explicitly acknowledged and appreciated by the two positive reviewers (CQsq and Bo1Q), who maintained scores **above the acceptance threshold** and highlighted the clarity, empirical rigor, and reasonable realism metrics of the revised version.

---

> ### Author Response · Authors · 2025-11-30
> **Global Response (2/2)**
>
> ### 2. Clarifying the evolution of the two initially negative reviews
>
> **Reviewer A99Y**
> Reviewer A99Y’s **initial review appears to have been partly based on a misreading or confusion with a different paper**: the summary describes a decentralized multi-agent fairness framework evaluated in Nocturne and Waymo Sim, which does not match our method, formulation, or experimental setup (CARLA, trajectory-level group setting, etc.). In our rebuttal we carefully re-explained:
>
> - the actual problem we address (AV-centric, route-conditioned CBV planning rather than generic cooperative MARL),
> - the role of the **group-relative term** (it operates over candidate trajectories of a single CBV, not across a population of agents),
> - and how we analyze failure cases.
>
> Once the reviewer understood the correct setting, they explicitly thanked us for the detailed clarifications and **raised their overall score to positive**, indicating that their main concerns came from the earlier misunderstanding rather than from fundamental flaws in the method.
>
> **Reviewer WoWw**
> Reviewer WoWw’s main concern is that our setup differs from the **Waymo Sim Agents Challenge (WOSAC)**, making it hard to judge RIFT’s realism metrics. In rebuttal we addressed this on several fronts:
>
> 1. **Problem setting difference (AV-centric vs unconditional sim agents).**
>    - We clarified that RIFT is **not a general-purpose sim-agent model** aimed at re-solving WOSAC. Instead, it is a **second-stage, AV-evaluation-oriented fine-tuning framework**: we start from an IL-pretrained model that already has strong unconditional realism, and then fine-tune it in a high-fidelity closed-loop simulator to obtain **AV-conditioned, interaction-controllable traffic** along specific routes and for specific AV planners.
>    - This AV-centric, route-conditioned objective is *not* captured by WOSAC’s offline, unconditional evaluation, so directly porting RIFT back into WOSAC after RLFT would not faithfully reflect its intended use case.
> 2. **Why CARLA/Bench2Drive, and how we align with WOSAC metrics.**
>    - We explained that CARLA/Bench2Drive is chosen because it supports **sensor-level, closed-loop evaluation of modern end-to-end planners (UniAD, VAD)**, which require rendering and control, something vectorized WOSAC environments cannot provide.
>    - Within CARLA, we **mirror WOSAC’s metric families** as faithfully as possible (kinematics, interaction, map adherence) and additionally introduced **JSD-based comparisons to WOMD** to provide a direct link back to real-world statistics. Under these metrics, RIFT maintains (or slightly improves) realism relative to the IL baseline and RLFT baselines.
> 3. **Cross-validation by other reviewers and prior literature.**
>    - The new distribution-matching experiments and realism analysis were explicitly **endorsed by CQsq and Bo1Q**, who considered our metrics reasonable and in line with prior closed-loop traffic-simulation works (e.g., NADE/NeuralNDE/RTR-style distribution-level evaluation).
>    - Due to time limits, Reviewer WoWw did not provide a further follow-up, but their main concerns about realism measurement and WOSAC comparability are now addressed in a way that is consistent with the broader literature and supported by the other reviewers.
>
> ------
>
> ### 3. Takeaway for the AC
>
> In summary, after rebuttal:
>
> - The two positive reviewers (CQsq, Bo1Q) maintain **above-threshold scores**, emphasizing clarity, empirical rigor, and reasonable realism/controllability metrics.
> - The initially negative reviewer **A99Y raised their score to positive** once the misunderstanding about our problem setting and experiments was resolved.
> - For Reviewer WoWw, we provided a comprehensive response: we clarified the conceptual gap between RIFT and WOSAC-style sim-agent benchmarks, added **WOSAC-aligned distributional realism metrics against WOMD**, and further analyzed our AV-evaluation protocol. These additions have been independently validated by the other reviewers.
>
> We hope this makes clear that the revised paper not only addresses the main concerns raised during review, but also provides a coherent and well-positioned contribution: **a dual-stage AV-centric traffic simulation framework that combines IL pre-training in a data-driven simulator with group-relative RL fine-tuning in CARLA, preserving trajectory-level realism while delivering interactive, controllable CBV behaviors that effectively reveal the weaknesses of modern learning-based planners under closed-loop evaluation.**

---

### Meta-Review · Area_Chair_GmAL · 2025-12-08

**Summary:**

This paper proposes RIFT, a two-stage framework that combines IL pre-training on nuPlan with group-relative RL fine-tuning in CARLA, freezing the trajectory generator and only adapting a scoring head to obtain “realistic and controllable” background traffic for end-to-end AV evaluation. Reviewers find the paper clearly written and the overall pipeline interesting, and acknowledge that the proposed GRPO/PPO-style fine-tuning is reasonably motivated for the “IL generator + scoring head” architecture. However, while I agree the paper itself presents some interesting directions worth exploration, both the algorithmic novelty and the empirical validation of the realism/controllability claims fall short of what I would expect for ICLR. I encourage the authors to submit it to a later venue.

**Reviewer Concerns:**

The rebuttal usefully clarified the setting, added WOMD-based distributional metrics, more covariate-shift analysis, and several ablations on CBV selection, which addressed some misunderstandings and convinced one initially negative reviewer to upgrade their score. That said, important concerns remain: the RL component is largely an incremental variant of existing GRPO/PPO-style methods; traffic “realism” is supported mainly by low-dimensional distributional distances and safety statistics without a more comprehensive analysis of multi-agent interaction patterns; and “controllability” is mostly realized as fixed reward shaping during training rather than clearly demonstrated test-time style control. As an AV evaluation framework, the experiments with a small set of planners and scenarios are a good first step, but they do not yet constitute a robust benchmark that clearly establishes the advantages of RIFT over stronger or more diverse baselines.

**Reviewer Scores:**

Two reviewers (CQsq, Bo1Q) are marginally positive, appreciating the clarity and engineering but also asking for stronger realism/style metrics and more representative baselines; A99Y raised their score after the rebuttal clarified the formulation and group-relative objective. The remaining reviewer (WoWw) maintains a strong reject, primarily due to concerns about the strength and positioning of the evaluation. While I do not fully share the strict requirement to align with specific state-space benchmarks, I agree that, given the incremental algorithmic novelty and the current limitations of the end-to-end evaluation, the overall contribution is below the ICLR bar, and I recommend rejection.

---

### Decision · Program_Chairs · 2026-01-26

Reject